# A multisynaptic pathway from the ventral midbrain toward spinal motoneurons in monkeys

Michiaki Suzuki[1,2,3] , Ken-ichi Inoue[4], Hiroshi Nakagawa[4] , Hiroaki Ishida[1], Kenta Kobayashi[5], Tadashi Isa[2,3,5,6,7] , Masahiko Takada[4] and Yukio Nishimura[1,2,3]

[1]*Neural Prosthetics Project, Tokyo Metropolitan Institute of Medical Science, Setagaya, Tokyo, Japan*
[2]*Department of Developmental Physiology, National Institute for Physiological Sciences, Okazaki, Aichi, Japan*
[3]*Department of Physiological Sciences, School of Life Science, SOKENDAI (The Graduate University for Advanced Studies), Hayama, Kanagawa, Japan*
[4]*Systems Neuroscience Section, Primate Research Institute, Kyoto University, Inuyama, Aichi, Japan*
[5]*Section of Viral Vector Development, National Institute for Physiological Sciences, Okazaki, Aichi, Japan*
[6]*Department of Neuroscience, Graduate School of Medicine, Kyoto University, Sakyo, Kyoto, Japan*
[7]*Institute for the Advanced Study of Human Biology (WPI-ASHBi), Kyoto University, Sakyo, Kyoto, Japan*

Edited by: Richard Carson & Jing-Ning Zhu

The peer review history is available in the Supporting Information section of this article (https://doi.org/10.1113/JP282429#support-information-section).

**Abstract** Motivation boosts motor performance. Activity of the ventral midbrain (VM), consisting of the ventral tegmental area (VTA), the substantia nigra pars compacta (SNc) and the retrorubral field (RRF), plays an important role in processing motivation. However, little is known about the neural substrate bridging the VM and the spinal motor output. We hypothesized that the VM might

**Michiaki Suzuki** is a researcher at the Neural Prosthetics Project, Tokyo Metropolitan Institute of Medical Science, Tokyo. He obtained his bachelor's degree in Physical Therapy at Tokyo Metropolitan University and PhD in Physiological Sciences at The Graduate University for Advanced Studies (SOKENDAI). His PhD research focused on the functional role of the mesolimbic system in motor control, using macaque monkeys. His current research interests focus on the development of a brain–machine interface to improve motor function through intervention in motivation-related brain areas.

exert a modulatory influence over the descending motor pathways. By retrograde transneuronal labelling with rabies virus, we demonstrated the existence of multisynaptic projections from the VM to the cervical enlargement in monkeys. The distribution pattern of spinal projection neurons in the VM exhibited a caudorostral gradient, in that the RRF and the caudal part of the SNc contained more retrogradely labelled neurons than the VTA and the rostral part of the SNc. Electrical stimulation of the VM induced muscle responses in the contralateral forelimb with a delay of a few milliseconds following the responses of the ipsilateral primary motor cortex (M1). The magnitude and number of evoked muscle responses were associated with the stimulus intensity and number of pulses. The muscle responses were diminished during M1 inactivation. Thus, the present study has identified a multisynaptic VM–spinal pathway that is mediated, at least in part, by the M1 and might play a pivotal role in modulatory control of the spinal motor output.

(Received 28 September 2021; accepted after revision 10 January 2022; first published online 5 February 2022)

**Corresponding author** Yukio Nishimura: Neural Prosthetics Project, Tokyo Metropolitan Institute of Medical Science, 2-1-6 Kamikitazawa, Setagaya, Tokyo 156-8506, Japan.    Email: nishimura-yk@igakuken.or.jp

**Abstract figure legend** We found a multisynaptic projection from the ventral midbrain (VM), consisting of the ventral tegmental area (VTA), substania nigra pars compacta (SNc) and retrorubral field (RRF), to the spinal cord in monkeys. There was a caudorostral gradient, in that the caudal VM (i.e. the caudal part of the SNc and RRF) contained more neurons projecting indirectly to the spinal cord than the rostral VM (i.e. the rostral part of the SNc and VTA). Activation of the VM induced responses in the primary motor cortex (M1) and several forelimb muscles. These muscle responses were mediated by M1. The VM might have a modulatory action on descending motor pathways.

## Key points

- Motivation to obtain reward is thought to boost motor performance, and activity in the ventral midbrain is important to the motivational process.
- Little is known about a neural substrate bridging the ventral midbrain and the spinal motor output.
- Retrograde trans-synaptic experiments revealed that the ventral midbrain projects multisynaptically to the spinal cord in macaque monkeys.
- Ventral midbrain activation by electrical stimulation generated cortical activity in the motor cortex and forelimb muscle activity.
- A multisynaptic ventral midbrain–spinal pathway most probably plays a pivotal role in modulatory control of the spinal motor output.

## Introduction

We experience that behavioural outcomes depend on our own motivation. For example, athletes are likely to produce their best performance in prestigious competitions, such as the Olympic games. Motivation to obtain a gold medal and/or social prestige might boost motor performance. Indeed, human studies have shown that motivational signals in relationship to high rewards enhance motor actions (i.e. force production; Pessiglione *et al.* 2007). In contrast, patients with major depression or Parkinson's disease, a symptom of which is apathy, exert less physical effort to obtain reward compared with heathy subjects (Cléry-Melin *et al.* 2011; Chong *et al.* 2015). Thus, motivation might be a key factor to boost motor performance. Activity of the primary motor cortex (M1), one of the origins of the descending motor pathway,

is modulated by external stimuli associated with reward (Marsh *et al.* 2015; Ramakrishnan *et al.* 2017; An *et al.* 2019). These results suggest that motor output derived from the descending motor pathway might be affected by motivational signals.

The activity of the ventral tegmental area (VTA) and the substantia nigra pars compacta (SNc), which predominantly contain dopaminergic (DA) neurons, plays an important role in processing motivation (Schultz, 1998; Bromberg-Martin *et al.* 2010). In addition to dopaminergic neurons, glutamatergic neurons serve as a reinforcer independent from dopamine to control reward-seeking behaviour (Zell *et al.* 2020). It is generally accepted that the ventral midbrain (VM), consisting of the VTA (termed A10), SNc (A9) and the retrorubral field (RRF; A8), has direct projections to widespread

**Table 1. Summary of animals used in neuroanatomical experiments**

| Monkey | Species | Sex | Body weight (kg) | Injection site | Survival time | Tracer | Injection tracks (*n*) | Injection volume (µl) |
|---|---|---|---|---|---|---|---|---|
| I | *Macaca fuscata* | Male | 4.5 | C6–C7 | 8 weeks | Biotinylated dextran amine | 6 | 3.6 |
| F | *Macaca mulatta* | Female | 5 | C6–T1 | 84 h | Rabies virus | 13 | 13 |
| L | *Macaca mulatta* | Female | 5 | C6–T1 | 90 h | Rabies virus | 13 | 15.6 |

**Table 2. Summary of animals used in electrophysiological experiments**

| Monkey | Species | Sex | Body weight (kg) | VM stimulation | EMG implant | ECoG implant | M1 chamber implant for muscimol injections |
|---|---|---|---|---|---|---|---|
| T | *Macaca mulatta* | Male | 8 | Left side | Right forelimb | Left side | — |
| D | *Macaca fuscata* | Female | 5.5 | Both sides | Both forelimbs | Left side | Right side |
| Y | *Macaca fuscata* | Female | 6.5 | Left side | Right forelimb | — | Left side |

Abbreviations: ECoG, electrocorticogram; M1, primary motor cortex; VM, ventral midbrain.

areas of the frontal cortex, including the M1, in primates (Gaspar *et al.* 1992; Williams & Goldman-Rakic, 1998; Zubair *et al.* 2021) and rodents (Hosp *et al.* 2011; 2015). Also, the VM can exert a modulatory impact on cortical activity, including that of the M1 (Arsenault *et al.* 2014; Kunori *et al.* 2014; Murris *et al.* 2020). However, it is unknown whether the VM can exert modulatory actions on spinal motor outputs via descending motor pathways.

In the present study, we hypothesized that the VM might give rise to a neural circuit for achieving motor actions. To address this issue, we investigated the existence of multisynaptic projections from the VM to the spinal cord by injecting rabies virus (RV), which permits retrograde trans-synaptic transport, into the cervical enlargement. Also, to explore the functional significance of the identified multisynaptic VM–spinal projections, muscle responses to electrical stimulation of the VM were recorded from the contralateral forelimb, coincidently with field responses in the ipsilateral M1. Finally, to elucidate the contribution of the M1 to muscle activity evoked by the VM stimulation, we compared the muscle responses before and during M1 inactivation. Our overall results have demonstrated a multisynaptic VM–spinal pathway that might underlie modulatory control of the descending motor output to forelimb muscles.

## Methods

### Animals and ethical approval

Six macaque monkeys (weighing 5.0–8.0 kg) were used for the present study (Tables 1 and 2). All the monkeys were supplied by the National BioResource Project or Kawahara Bird-Animal Trading Co. We performed neuroanatomical and electrophysiological experiments described in the sections below (neuroanatomical experiments and electrophysiological experiments). All animal experiments were approved by the committees for animal experimentation of the Primate Research Institute, Kyoto University (approval numbers 2015-050, 2018-062 and 2019-053), the National Institutes of Natural Sciences (approval number 15A079) and the Tokyo Metropolitan Institute of Medical Science (approval numbers 18034, 19049 and 20-052). All experimental procedures were conducted in accordance with the US National Institutes of Health *Guide for the Care and Use of Laboratory Animals, 8th edition*. Throughout the experiments, the monkeys were housed in individual cages at an ambient temperature of 23–26°C and a 12 h–12 h on–off light cycle. The animals were fed regularly with diet pellets and had free access to water. They were monitored closely, and animal welfare was assessed on a daily basis or, if necessary, several times a day.

### Neuroanatomical experiments

**Animals.** Three macaque monkeys (monkeys I, F and L) were used for the neuroanatomical experiments (Table 1). Monkey I received injections of biotinylated dextran amine (BDA), which permits retrograde labelling, to investigate direct projections from the VM to the spinal cord. Monkeys F and L received injections of the retrograde trans-synaptic tracer, RV, to investigate trans-synaptic projection. Experiments involving RV were performed in a special primate laboratory (biosafety level

2) designated for *in vivo* infectious experiments at the Primate Research Institute, Kyoto University. Throughout the experiments, the monkeys that received RV injections were housed in individual cages that were installed inside a special biosafety cabinet.

**Preparation of BDA and RV.** A 20% solution of BDA (3000 molecular weight; Molecular Probes, Eugene, OR, USA) in 0.01 M phosphate buffer (PB; pH 7.4), as a conventional retrograde tracer, and the challenge-virus-standard (CVS-11) strain of RV, which exhibits retrograde trans-synaptic transport, were used in the present study. The RV was originally obtained from the Centers for Disease Control and Prevention (Atlanta, GA, USA) and was donated by Dr Satoshi Inoue (The National Institute of Infectious Diseases, Tokyo, Japan). The viral batch used in the present study was the same as that used in a previous study (Ishida *et al.* 2016). The titre of the viral suspension was $1.0 \times 10^8$ focus-forming units (FFU)/ml.

**Surgery for injections into the spinal cord.** The procedures described below were performed under general anaesthesia induced by ketamine (Daiichi-Sankyo, Japan, 10 mg/kg, I.M.) plus xylazine (Bayer, German, 1 mg/kg, I.M.) and maintained with sodium pentobarbital (Kyoritsu Seiyaku, Japan, 20 mg/kg, I.V.) or 1–1.5% isoflurane (MSD Animal Health, Japan). The depth of anaesthesia was confirmed by pain response. During surgery, the vital signs of the animal, such as respiratory and circulatory parameters (respiratory rate, inspiratory carbon dioxide concentration, percutaneous oxygen saturation and heart rate) and body temperature, were monitored carefully. There was no evidence of tachy-cardia or tachypnoea during surgical procedures, nor major deviation in the heart rate or respiratory rate in response to noxious stimuli. The absence of reflexive movements to noxious stimuli and corneal reflex was also used to verify the level of anaesthesia.

All surgical procedures were performed in sterile conditions. The skin and axial muscles were dissected at the level of the C3 to T2 vertebrae. Subsequently, laminectomy was performed to expose the cervical segments of the spinal cord. With the dura mater open, dorsal root entry zones were used to identify segmental levels. A 10 $\mu$l Hamilton microsyringe needle was tilted to enter the spinal cord at an angle. The angle and depth of the syringe required to reach the intermediate zone and ventral horn of the spinal cord were adjusted for each segment. Repetitive penetrations spaced 2 mm apart were made rostrocaudally from the C6 to C7 spinal segments for BDA injections and from the C6 to T1 spinal segments for RV injections. For BDA injection, a total of 3.6 $\mu$l (monkey I; BDA, 0.3 $\mu$l, two depths and six tracks) of

the BDA solution was injected into the spinal cord. For RV injection, a total of 13 $\mu$l (monkey F; RV, 0.5 $\mu$l, two depths and 13 tracks) or 15.6 $\mu$l (monkey L; RV, 0.6 $\mu$l, two depths and 13 tracks) of the viral suspension was injected into the spinal cord. After all injections were completed, the muscle and skin were sutured with nylon or silk threads.

Postoperative management consisted of observation of the animals until they had recovered completely from the general anaesthesia. Dexamethasone (Sandoz Pharma K.K., Japan, 0.825 mg/kg, I.M.), ampicillin (Meiji Seika Pharma Co., Ltd., Japan, 40 mg/kg, I.M.) and ketoprofen (Kissei Pharmaceutical Co., Ltd., Japan, 2.0 mg/kg, I.M.) were administrated twice (morning and evening) per day until postoperative day 3. Daily observation of the animals for evaluation of their health status was conducted by veterinarians and the researchers. The survival time after injection was set at 8 weeks (monkey I) for BDA and at 3.5 days (monkey F) or 3.75 days (monkey L) for RV (Table 1).

**Histological procedures.** After the survival periods, the monkeys were deeply anaesthetized with an overdose of sodium pentobarbital (50 mg/kg, I.V.). After confirmation of loss of pain reflex under deep general anaesthesia, the monkeys were euthanized by transcardial perfusion with 0.1 M PBS, followed by 10% formalin dissolved in 0.1 M phosphate buffer. Death of the animals was confirmed by respiratory arrest, cardiac arrest and pupillary dilatation. The fixed brain and spinal cord were harvested, post-fixed in the same fresh fixative overnight at 4°C, and equilibrated with 30% sucrose in 0.1 M phsophate buffer at 4°C.

Coronal sections were cut serially at 50 $\mu$m thickness on a freezing microtome. Every 10th section was mounted onto gelatin-coated slides and Nissl stained with 1% Cresyl Violet. The remaining series of the sections at 500 $\mu$m apart were processed for immunohistochemistry to visualize the BDA or RV labelling and dopaminergic cell groups.

For visualization of injected and transported BDA, the sections were pretreated with 1% $H_2O_2$ for 30 min and washed three times in 0.1 M PBS. The sections were then incubated in 0.1 M PBS containing 0.4% Triton X-100 overnight, followed by fresh PBS containing avidin–biotin–peroxidase complex (ABC Elite: 1:100 dilution; Vector Laboratories, USA) in 0.1 M PBS for 3 h at room temperature. Subsequently, the sections were reacted for 10–20 min in 0.05 M Tris–HCl buffer (pH 7.6) containing 0.04% diaminobenzene tetrahydrochloride (Wako, Japan), 0.04% $NiCl_2$ and 0.003% $H_2O_2$. These sections were counterstained with 0.5% Neutral Red, mounted onto gelatin-coated glass slides, dehydrated and coverslipped.

For immunoperoxidase staining for RV, the sections were pretreated with 0.3% $H_2O_2$ for 30 min, washed three times in 0.1 M PBS and immersed in 1% skimmed milk for 1 h. Subsequently, the sections were incubated for 2 days at 4°C with rabbit anti-RV antibody (donated by Dr S. Inoue in National Institute of Infectious Diseases, Tokyo, Japan) in 0.1 M PBS containing 2% normal donkey serum and 0.1% Triton X-100. The sections were then incubated with biotinylated donkey anti-rabbit IgG antibody (1:1000 dilution; Jackson Laboratories, USA) in the same fresh medium for 2 h at room temperature, followed by the avidin–biotin–peroxidase complex kit (ABC Elite; 1:200 dilution; Vector Laboratories, USA) in 0.1 M PBS for 2 h at room temperature. To visualize the antigen, the sections were reacted for 10–20 min in 0.05 M Tris–HCl buffer (pH 7.6) containing 0.04% diaminobenzene tetrahydrochloride (Wako, Japan), 0.04% $NiCl_2$ and 0.002% $H_2O_2$. These sections were counterstained with 0.5% Neutral Red, mounted onto gelatin-coated glass slides, dehydrated and coverslipped.

For immunoperoxidase staining for dopaminergic neurons, the same procedure as for immunoperoxidase staining for RV, as described above, was performed. A mouse monoclonal anti-tyrosine hydroxylase (TH) antibody (1:500 dilution; Millipore) was used as the primary antibody, and a biotinylated donkey anti-mouse IgG antibody (1:1000 dilution; Jackson Laboratories, USA) was used as a secondary antibody. The reacted sections were mounted onto gelatin-coated glass slides, dehydrated and coverslipped.

For double immunofluorescence histochemistry for anti-RV and anti-TH, the sections were rinsed three times in 0.1 M PBS, immersed in 1% skimmed milk for 1 h, then incubated with rabbit anti-RV antibody and mouse monoclonal anti-TH antibody (1:500 dilution; Millipore). The sections were then incubated for 2 h at room temperature with a cocktail of Alexa 488-conjugated donkey anti-rabbit IgG antibody (1:400 dilution; Jackson Laboratories) and Cy3-conjugated donkey anti-mouse IgG antibody (1:400 dilution; Jackson Laboratories) in the same fresh incubation medium. These sections were mounted onto gelatin-coated glass slides and coverslipped.

**Delineation of dopaminergic cell groups in the VM.** Sections stained for TH immunoreactivity were used to estimate the boundaries of areas for the ventral midbrain dopamine (DA) cell groups, VTA, SNc and RRF (Dahlstrom & Fuxe, 1964). The virtual absence of noradrenergic neurons in the midbrain renders TH a specific marker for dopaminergic cells in the mesencephalon (Arsenault *et al.* 1988; Deutch *et al.* 1988; Gaspar *et al.* 1992).

**Data analysis for labelled neurons.** Neuronal labelling was plotted with Neurolucida (MicroBrightField, Williston, VT, USA) on tracing of equidistant coronal sections (500 $\mu$m) through VM (immunoperoxidase staining for BDA, 16 sections in monkey I; immunoperoxidase staining for RV, 15 sections in monkey F and 16 sections in monkey L; double immunofluorescence histochemistry for RV and TH, 14 sections in monkey F and 15 sections in monkey L). Labelled neurons in the VTA, SNc and RRF were counted with Neurolucida. Photomicrographs were captured to display labelled neurons in the regions of interest.

## Electrophysiological experiments

**Animals.** To clarify the functional connectivity between the VM and muscles, three adult macaque monkeys (monkeys T, D and Y) were used for a series of electrophysiological experiments (Table 2). We performed two types of electrophysiological experiments: one was recording cortical responses and muscle responses induced by VM stimulation (monkeys D and T); the other was recording muscle responses before and during M1 inactivation (monkeys D and Y). All three monkeys were chronically implanted with a chamber for VM stimulation and microwires into multiple upper limb muscles for EMG recording. Monkeys D and T were also chronically implanted with electrocorticogram (ECoG) arrays for recording cortical responses to VM stimulation. In addition, monkeys D and Y were implanted with an additional chamber for the M1 inactivation study.

**Surgery for chronic implants.** General anaesthesia was induced and monitored as for the spinal cord injections. All surgical procedures were performed in sterile conditions.

*Implantation of a chamber for VM stimulation.* Magnetic resonance imaging (MRI) was used to determine the precise geometry of the VM to allow accurate placement of microelectrode penetrations for VM stimulation. Scans were carried out under deep anaesthesia, which was introduced with ketamine (10 mg/kg, I.M.) plus xylazine (1 mg/kg, I.M.) and maintained with sodium pentobarbital (20 mg/kg, I.M.). The depth of anaesthesia was confirmed by pain response. T1-weighted images were collected with a 3 T scanner (Allegra, Siemens, Germany). In three monkeys (monkeys D, T and Y), surgery was performed to gain easy access for electrical stimulation of the VM. After partial craniotomy, in which the centre was located above the VM, a custom-made delrin chamber was attached to cover the craniotomy at an angle of 10° to the mid-sagittal plane (monkeys D and T) or without tilting (monkey Y).

Two titanium–steel tubes were mounted in parallel over the frontal and occipital lobes for fixation of the head. The chamber and titanium tubes were fixed to the screws with acrylic resin.

Postoperative management consisted of observation of the operated animals until they had recovered completely from the anaesthesia, and administration of ampicillin (40 mg/kg, I.M.) and ketoprofen (2.0 mg/kg, I.M.). Daily observations of the animals for evaluation of their health status were conducted by veterinarians and researchers.

*Implantation of microwires for EMG recordings.* Pairs of Teflon-insulated wire electrodes (AS631; Cooner Wire), which were tunnelled subcutaneously to their target muscles, were secured into the forelimb muscles on the contralateral side to the VM stimulation side using silk sutures. A connector (Samtec, Singapore) was fixed to the skull with acrylic resin. In the right forelimb muscles of monkeys T and D, the wire electrodes were implanted in one shoulder muscle [pectoralis major (PEC)], two elbow muscles [triceps brachii (TRI) and biceps brachii (BB)], five wrist muscles [extensor carpi radialis (ECR), extensor carpi ulnaris (ECU), flexor carpi radialis (FCR), flexor carpi ulnaris (FCU) and palmaris longus (PL)], four digit muscles [extensor digitorum communis (EDC), extensor digitorum 2,3 (ED23), flexor digitorum superficialis (FDS) and flexor digitorum profundus (FDP)] and one intrinsic hand muscle [first dorsal interosseous (FDI)]. In the left forelimb muscles of monkey D, the wire electrodes were implanted in one shoulder muscle [deltoid (DEL)], one elbow muscle (TRI), five wrist muscles (ECR, ECU, FCR, FCU and PL), three digit muscles [EDC, extensor digitorum 4, 5 (ED45) and FDS] and one intrinsic hand muscle [adductor pollicis (ADP)]. In the right forelimb muscles of monkey Y, the wire electrodes were implanted in one shoulder muscle (DEL), two elbow muscles (TRI and BB), five wrist muscles (ECR, ECU, FCR, FCU and PL), three digit muscles (EDC, ED45 and FDS) and one intrinsic hand muscle (FDI). The implanted muscles were identified by stimulation through each electrode pair and observing the evoked movements during the surgery.

Postoperative management consisted of observation of the operated animal until it had recovered completely from general anaesthesia, and administration of ampicillin (40 mg/kg, I.M.), and ketoprofen (2.0 mg/kg, I.M.). Daily observations of the animals for evaluation of their health status were conducted by veterinarians and researchers.

*Implantation of ECoG arrays for cortical activity recordings.* To implant a grid electrode array on the cortical surface of the M1, the cortices around the arcuate sulcus and the central sulcus on the left side were expose by a craniotomy. In monkeys D and T, we implanted a 21-channel grid electrode array (3 × 7), in which the diameter of each electrode was 1 mm and the inter-electrode distance was 5 mm (Unique Medical, Japan), on the digit, wrist and arm areas of the M1 beneath the dura mater. We placed the earth and reference electrodes over the ECoG electrode such that they contacted the dura. After implanting the array, the opening of the skull was covered, and a connector was fixed to the skull with acrylic resin.

Postoperative management consisted of observation of the operated animal until it had completely recovered from anaesthesia, and administration of dexamethasone (0.825 mg/kg, I.M.), ampicillin (40 mg/kg, I.M.) and ketoprofen (2.0 mg/kg, I.M.). Daily observations of the animals for evaluation of their health status were conducted by veterinarians and researchers.

*Implantation of a chamber for M1 inactivation.* The cortex around the central sulcus was exposed unilaterally (right side for monkey D and left side for monkey Y) by craniotomy, and a custom-made delrin chamber was attached to cover the craniotomy and fixed to the skull with acrylic resin.

Postoperative management consisted of observation of the operated animal until it had recovered completely from anaesthesia, and administration of ampicillin (40 mg/kg, I.M.) and ketoprofen (2.0 mg/kg, I.M.). Daily observations of the animals for evaluation of their health status were conducted by veterinarians and researchers.

**Electrical stimulation of the VM.** Before data collection, the monkeys were trained to sit in a primate chair. After completion of chair training, experiments were conducted. The monkeys were sedated with ketamine (10 mg/kg, I.M.), and their heads were fixed to the primate chair. The depth of anaesthesia was confirmed by pain response. Additional doses of ketamine were given as needed to eliminate spontaneous movements during the data-recording sessions.

Stimulation sites in the VM were determined stereotaxically based on the MRI image. Stimuli were delivered using bipolar concentric tungsten, platinum or stainless-steel electrodes (impedance, 500–600 kΩ; Unique Medical, Japan). Electrodes were positioned in the recording chamber with an $x$–$y$ grid and guided through the cerebrum via a stainless-steel cannula using an $x$–$y$ coordinate manipulator mounted on the head chamber. An electrode was inserted into the unilateral VM using a manual hydraulic microdrive. Stimulus trains with a current of 100, 300 or 500 $\mu$A were delivered to the target location >15 times through a constant-current stimulator. Stimuli consisted of a single biphasic pulse (a positive phase followed by a negative phase) or three biphasic pulses, with a 0.2 ms square-wave duration at a frequency of 300 Hz. For the M1 inactivation study,

three-pulse stimulation with a current of 300 $\mu$A was delivered to the VM. Stimulus trains were separated by >470 ms. Eye movements evoked by VM stimulation were used as a landmark to ensure electrode position, because the oculomotor nerve root passes around the VTA.

**M1 inactivation.** To determine the injection sites in monkeys D and Y, somatotopic maps were investigated using intracortical microstimulation under sedation with ketamine (10 mg/kg, I.M.), as described in our previous study (Nishimura *et al.* 2007). The depth of anaesthesia was confirmed by pain response. A tungsten microelectrode (1.1–1.5 M$\Omega$ at 1 kHz) was inserted perpendicular to the cortical surface using a hydraulic micromanipulator. Regions in the precentral gyrus were mapped with intracortical microstimulation. Stimuli consisted of 15 biphasic pulses (a positive phase followed by a negative phase), with a 0.2 ms square-wave duration at a frequency of 333 Hz delivered using a constant-current stimulator. Evoked movements of various body parts were observed carefully and detected by visual inspection and direct muscle palpation. When no movement was observed at a current intensity of 50 $\mu$A, the stimulation site was considered a no-response site.

According to the results of intracortical micro-stimulation, muscimol, a $\gamma$-aminobutyric acid type A (GABA$_A$) receptor agonist (5 $\mu$g/$\mu$l, dissolved in 0.1 M phosphate buffer, pH 7.4), was slowly injected (five or six sites, 1 $\mu$l/site) by pressure at a rate of 0.2 $\mu$l/min through a stainless-steel microinjection needle connected to a 10 $\mu$l Hamilton microsyringe (Hamilton Company, USA) via a microtube to inactivate M1 activity. A total of 5 or 6 $\mu$l of muscimol solution was injected into M1 digit, wrist and shoulder areas in monkey D and into M1 digit and wrist areas in monkey Y. The depth with the lowest motor threshold was chosen as the injection site. As a control, the same volume saline was injected at the same sites as muscimol.

**Data collection.** Time stamps were used to record the timing of the stimulation. The ECoG signals from ipsilateral cortices and/or EMG signals from the contralateral side elicited by the VM stimulation were recorded simultaneously using a Cerebus data acquisition system (Blackrock Microsystems, Salt Lake City, UT, USA) at a sampling rate of 2000 Hz. The ECoG and/or EMG signals were extracted using multichannel amplifiers with a bandpass analog filter (0.3 Hz high-pass and 7500 Hz low-pass) and with bandpass digital filters in the Neuronal Signal Processor (2000 Hz low-pass for ECoG signals and 30–1000 Hz bandpass for EMG signals). Data containing apparent noise were not used in analyses.

**Procedure for stimulus-triggered averages.** Stimulus-triggered averages of ECoG signals obtained from the M1 and rectified EMG signals obtained from forelimb muscles were constructed using software custom written in MATLAB 2012b (MathWorks, Natick, MA, USA). Averages were compiled over a 300 ms epoch that included 100 ms before and 200 ms after the first stimulus trigger. To detrend the baseline activity, we subtracted the mean baseline activity (from −100 to 0 ms) from the averaged data.

Processed data were normalized by standard deviation (SD) of the baseline for each signal and expressed in multiples of the SD of the baseline (signal-to-noise ratio; Cheney *et al.* 1991). The M1 response was identified as having a significant response when the evoked response exceeded ±2SD of the baseline between 0 and 200 ms after stimulus onset. Each muscle was considered to have a significant activation when the magnitude of the stimulus-triggered average exceeded +2SD of the baseline and had a total duration ≥3 ms between 5 and 25 ms after stimulus onset.

**Quantification of muscle responses.** The magnitude of the stimulus-triggered average was quantified by the mean percentage increase (MPI) of the feature above baseline (Fetz & Cheney, 1980; Nishimura *et al.* 2013). The feature was determined by the onset and offset of the muscle response. Onset and offset of the evoked muscle responses were defined as the initial point and final point in the first period during which the data points continuously exceeded +2SD, respectively. Data for which it was not possible to identify the onset because of stimulus artefacts were excluded from this analysis.

**Histological confirmation of simulation sites.** At the end of the experiment on each animal, the monkeys were deeply anaesthetized with an overdose of sodium pentobarbital (50 mg/kg, I.V.) to carry out electro-coagulation for identification of stimulation sites in the VM. After confirmation of loss of pain reflex under deep general anaesthesia, electrocoagulation was carried out using a rectangular constant current at 30 $\mu$A for 20 s through the stimulating electrode. Then, the monkeys were euthanized by transcardial perfusion with 0.1 M PBS, followed by 10% formalin dissolved in 0.1 M phosphate buffer. Death of the animals was confirmed by respiratory arrest, cardiac arrest and pupillary dilatation.

The perfused brain was harvested, postfixed in the same fresh fixative overnight at 4°C, and saturated with 10% sucrose in 0.1 M phosphate buffer (pH 7.3), followed by saturation in 20 and 30% sucrose solutions. The perfused brain was cut serially into 50 $\mu$m coronal sections with a freezing microtome, and every fifth section was mounted

onto a gelatin-coated glass slide and Nissl stained with 0.1% Cresyl Violet.

To delineate boundaries of dopaminergic cell groups in the VM, anti-TH immunohistochemistry was also performed for the remaining sections in monkeys D and Y, as described in the 'Histological procedures' section above. Photomicrographs of the stimulation sites were captured, and then reconstructed. Electrical stimulation sites were estimated based on the location of coagulation in coronal sections of the brain. In monkey T, the boundaries of areas for the VTA, SNc and RRF in the VM were set based on the cytoarchitectonic structures by reference to the stereotaxic atlas (Paxinos *et al.* 2009).

### Statistical analysis

All statistical analyses were performed in MATLAB 2016b (MathWorks).

To compare the effect of VM stimulation with different current intensities (100, 300 and 500 $\mu$A) and the difference in the magnitude of muscle responses depending on the stimulation sites, we compared the MPI among three different stimulus currents, numbers or the three regions (VTA, SNc and RRF) using one-way ANOVA with repeated measures (Kruskal–Wallis test). *Post hoc* multiple comparisons were conducted using Bonferroni's correction. Statistical significance was defined as $P < 0.05$.

To confirm the effect of M1 inactivation on the muscle responses evoked by VM stimulation, we compared the MPI before and after microinjection of muscimol or saline (Student's paired *t* test, $P < 0.05$). Before this comparison, we accepted data showing no significant change (Student's paired *t* test, $P > 0.05$) in MPI between the initial period of a session (stimulus counts 1–250) and the last period (stimulus counts 751–1000) in both conditions before and after microinjection. For accepted data, we compared the feature duration ($>+2SD$) before with after micro-injection and adopted the longer feature duration for MPI calculation. Statistical significance was defined as $P < 0.05$.

## Results

### Retrogradely labelled neurons in the VM

To investigate the neuroanatomical connectivity from the VM to the spinal cord, three monkeys (monkeys I, F and L) were assigned to neuroanatomical experiments (Table 1).

First, to verify the possibility of a direct projection from the VM to the cervical spinal cord that innervates forelimb muscles, one monkey (monkey I) received injections of 20% BDA into the cervical enlargement (i.e. C6–C7; Fig. 1*A*). The unilateral injections were aimed at laminae VII and IX (Fig. 1*B*), where spinal interneurons and motoneurons innervating the upper limb muscles (Rexed, 1952; Jenny & Inukai, 1983) are distributed, respectively. Large layer V pyramidal neurons in the motor-related areas of the frontal lobe (i.e. corticospinal neurons; He *et al.* 1993; 1995) and large neurons in the red nucleus (RN; Kennedy *et al.* 1986; Holstege *et al.* 1988; Ralston *et al.* 1988; Burman *et al.* 2000) make mono-synaptic connections with the spinal interneurons and motoneurons. Therefore, we examined neuronal labelling not only in the VM, including the VTA, SNc and RRF, but also in the cortical motor-related areas and the RN. Numbers of BDA-labelled neurons were observed in layer V of the motor-related areas, including the caudal part of the dorsal premotor cortex (PMd) and the supplementary motor area (SMA), and the primary somatosensory cortex (S1), with the highest density in layer V of the M1 (Fig. 1*C* and *D*) and in the RN (Fig. 1*E*), but no BDA-labelled neurons were observed in the VM (Figs 1*E* and 2*A*). These results indicate that the VM has no direct projection to the cervical enlargement.

Second, to explore the existence of multisynaptic VM–spinal projections, two monkeys (monkeys F and L) received injections of the retrograde transneuronal tracer RV into the cervical enlargement (C6–T1; Fig. 1*F*). The RV injections were performed in the same manner as for the BDA injections (Fig. 1*G*), and these monkeys were allowed to survive for 84 h (monkey F) and 90 h (monkey L), which enables labelling of neurons at least second-order (disynaptic) from the spinal cord. Rabies virus-labelled neurons in the M1 (presumably, its forelimb region) were observed in layers III (Fig. 1*J*) and V (Fig. 1*I*), which is consistent with a previous report that M1 neurons in layer III became infected through corticomotoneuronal cells in layer V after RV injections into forelimb muscles (Rathelot & Strick, 2006). In the non-primary motor-related areas, such as the caudal part of the PMd and SMA, which project not only to the M1 (Barbas & Pandya, 1987; Luppino *et al.* 1993; Miyachi *et al.* 2005), but also to the cervical enlargement (He *et al.* 1993, 1995), RV-labelled neurons were found in layers III and V. However, the caudal part of the ventral premotor cortex (PMv) exhibited far fewer labelled neurons. The number of RV-labelled neurons in the M1 and non-primary motor-related areas was larger in the animal (monkey L) with a longer survival time.

Third, to confirm whether RV infection might reach third-order neurons (trisynaptically connected from the cervical enlargement), we investigated the RN–cerebellar circuit by RV labelling of Purkinje cells via the anterior interpositus nucleus (AIP; Fig. 3*A*; Kennedy *et al.* 1986; Voogd, 2014; Basile *et al.* 2021). Rabies virus labelling of AIP neurons was observed at survival times of 84 and 90 h (Fig. 3*B*–*E*). In addition, some Purkinje cells were labelled with RV at the 90 h survival time point (Fig. 3*F* and *G*). These results suggest that RV infection of the cervical

spinal cord reached second- and third-order neurons at 84 and 90 h survival times, respectively.

At the midbrain level, RV-labelled neurons were observed not only in the RN (Fig. 1*K*), but also in the VM, throughout the VTA, SNc and RRF (Fig. 1*L* and *M*) in both monkeys. Such neuronal labelling occurred bilaterally [6 neurons in contralateral VTA, 8 in ipsilateral VTA, 18 in contralateral SNc, 3 in ipsilateral SNc, 27 in contralateral RRF and 20 in ipsilateral RRF (monkey F); and 68 neurons in contralateral VTA, 62 in ipsilateral VTA, 63 in contralateral SNc, 63 in ipsilateral SNc, 94 in contralateral RRF and 108 in ipsilateral RRF (monkey L)]. Although these labelled neurons in the VM were widely distributed throughout its entire rostrocaudal extent, there was a caudorostral gradient, in that the caudal VM (i.e. the caudal part of the SNc and RRF) contained more labelled neurons than the rostral VM (i.e. the rostral part of the SNc and VTA; Fig. 2*B–D*). The above results indicate that

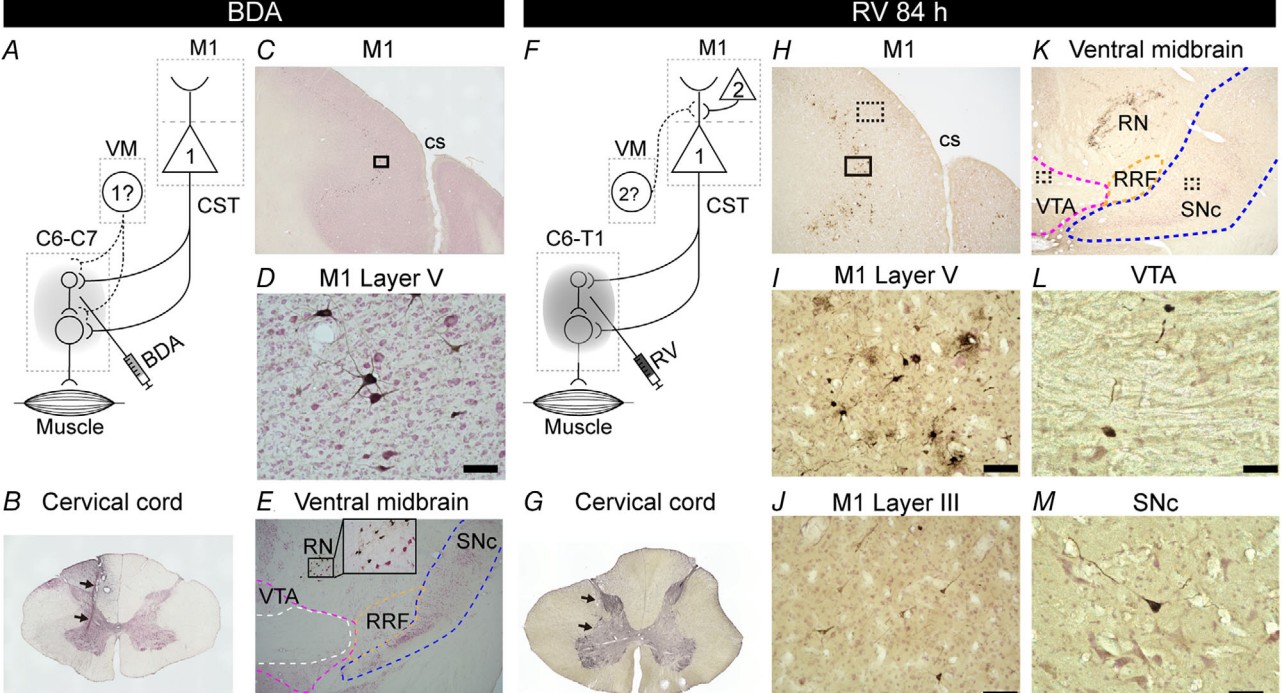

**Figure 1. Retrograde neuronal labelling from the cervical enlargement in the primary motor cortex and the ventral midbrain**
*A–E*, retrograde neuronal labelling with BDA injected into the cervical enlargement. *A*, diagram of retrograde transport of BDA from the cervical enlargement. The first-order neurons are specified by '1'. *B*, representative image of the injection site in the spinal cord at the level of C6. Arrows indicate the injection track. *C*, photomicrograph of a coronal section of the M1 and S1 contralateral to the BDA injection. *D*, higher magnification of the labelled neurons in M1 layer V. Scale bar, 100 $\mu$m. The magnified area corresponds to the black box in *C*. *E*, no labelled neurons in the VM and labelled neurons in the RN (inset, higher magnification of labelled RN neurons). Locations of the VTA (magenta), SNc (blue) and RRF (orange) are outlined based on the results of tyrosine hydroxylase immunohistochemistry. Data in *B–E* were obtained from monkey I. *F–M*, retrograde transneuronal labelling with RV injected into the cervical enlargement. *F*, diagram of retrograde trans-synaptic transport of RV from the cervical enlargement. The second-order neurons that connect to the first-order neurons are specified by '2'. *G*, representative image of the injection site in the spinal cord at the level of C7. Arrows indicate the injection track. *H*, photomicrograph of the coronal section including the contralateral M1 and S1 to the RV injection. *I*, higher magnification photomicrograph of the labelled neurons in M1 layer V. Scale bar, 100 $\mu$m. The magnified area corresponds to the black box in *H*. *J*, higher magnification photomicrograph of the labelled neurons in M1 layer III. Scale bar, 100 $\mu$m. The magnified area corresponds to the dotted black box in *H*. *K*, photomicrograph of a coronal section including the VTA, SNc, RRF and RN. *L*, higher magnification photomicrograph of the labelled neurons in the VTA by RV immunohistochemistry. Scale bar, 50 $\mu$m. The magnified area corresponds to the dotted black box in the VTA in *K*. *M*, higher magnification photomicrograph of the labelled neuron in the SNc by RV immunohistochemistry. Scale bar, 50 $\mu$m. The magnified area corresponds to the dotted black box in the SNc in *K*. Data in *G–M* were obtained from monkey F (survival time, 84 h). Abbreviations: BDA, biotinylated dextran amine; cs, central sulcus; CST, corticospinal tract; D, dorsal; L, lateral; M1, primary motor cortex; RN, red nucleus; RRF, retrorubral field; RV, rabies virus; S1, primary somatosensory cortex; SNc, substantia nigra pars compacta; VM, ventral midbrain; VTA, ventral tegmental area.

the VM has multisynaptic (at least disynaptic) projections to the cervical enlargement.

When double immunofluorescence histochemistry for RV and TH was performed, we found that only a small population of RV-labelled neurons displayed TH immunoreactivity [3.8% (3/79 neurons) in monkey F and 2.6% (10/391) in monkey L; Fig. 4]: for monkey F, 0% (0/19) in the VTA, 0% (0/11) in the SNc and 6.1% (3/49) in the RRF; and for monkey L, 2.0% (2/101) in the VTA, 4.4% (6/136) in the SNc and 1.3% (2/154) in the RRF. Given that RV interferes with protein synthesis in infected neurons (Miyachi *et al.* 2006), it should be noted that this result probably underestimates the number of double-labelled neurons.

### Ventral midbrain stimulation-evoked M1 and muscle responses

To investigate the functional significance of the anatomically identified multisynaptic VM–spinal

projections, two monkeys (monkeys D and T) were assigned to electrophysiological experiments (Table 2). Electrical stimulation with different current intensities (100, 300 or 500 $\mu$A) was delivered to the unilateral VM (31 sites in monkey D and five sites in monkey T) under sedation with ketamine (10 mg/kg, I.M.). Stimuli consisted of one or three biphasic pulses with 0.2 ms square-wave duration at a frequency of 300 Hz. Evoked responses from the ipsilateral M1 and the contralateral forelimb muscles to the VM stimulation were recorded simultaneously with ECoG and EMG, respectively (Fig. 5*A* and *B*). Stimulus-triggered averages of ECoG signals obtained from the M1 and rectified EMG signals were compiled for each stimulation site.

The single-pulse VM stimulation evoked short-latency negative responses of the M1 ($\sim$5 ms) and induced responses in a few muscles with some delay after the M1 responses. The M1 and muscle responses to the three-pulse stimulation were larger than those to the single-pulse stimulation (Fig. 5*C* and *D*). Although significant M1 responses were induced at all stimulation

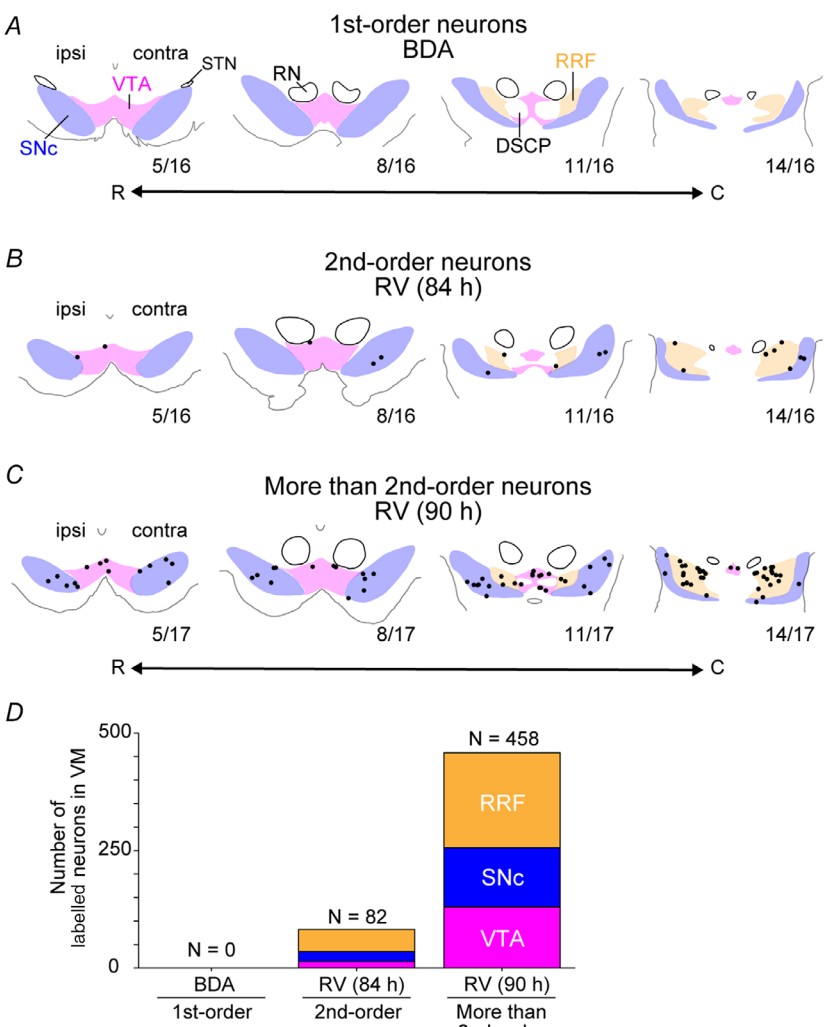

**Figure 2. Distribution patterns of retrogradely labelled neurons in the ventral midbrain after injections of biotinylated dextran amine or rabies virus into the cervical enlargement**

*A* and *B*, representative coronal sections throughout the rostrocaudal extent of the VM. Coloured areas indicate the regions of the VTA (magenta), SNc (blue) and RRF (orange), which were determined by tyrosine hydroxylase immunohistochemistry. Black dots indicate retrogradely labelled neurons. Only labelled neurons in the VM (VTA, SNc and RRF) are plotted. The rostrocaudal location of each section is indicated below the section. *A*, first-order neurons labelled with biotinylated dextran amine injected into the cervical enlargement. No labelled neurons are observed in the VM. Data were obtained from monkey I. *B*, second-order neurons with RV injected into the cervical enlargement. Data were obtained from monkey F (survival time, 84 h). *C*, more than second-order neurons with RV injected into the cervical enlargement. Data were obtained from monkey L (survival time, 90 h). *D*, distribution of labelled neurons in the VM, including the VTA, SNc and RRF. Abbreviations: C, caudal; contra, contralateral to the injection; DSCP, decussation of superior cerebellar peduncles; ipsi, ipsilateral to the injection; R, rostral; RN, red nucleus; RRF, retrorubral field; RV, rabies virus; SNc, substantia nigra pars compacta; STN, subthalamic nucleus; VM, ventral midbrain; VTA, ventral tegmental area.

sites for 100, 300 and 500 $\mu$A intensities, effective VM sites that elicited significant muscle responses were increased with stimulus intensity (Table 3). As shown in Fig. 5*D*, VM stimulation induced responses not only from proximal muscles, but also from distal muscles in the contralateral forelimb. The magnitude of the stimulus-triggered averages obtained from multiple forelimb muscles was quantified by the MPI between the onset and the offset of muscle responses. The MPI and the number of significantly evoked muscle responses increased with the stimulus intensity (Fig. 5*E* and *F*). Furthermore, the onset latencies of forelimb muscle responses (a total of 364 responses) were distributed within the range of 5.0–25.0 ms (Table 4). Comparing the distributions of onset latency between the three-pulse and the single-pulse stimulation showed that the three-pulse stimulation contained more slow-onset components.

To investigate the difference in the magnitude of muscle responses among the VTA, SNc and RRF, the MPI in each stimulus parameter was compared. In the higher-intensity conditions (500 $\mu$A $\times$ 1 and 500 $\mu$A $\times$ 3), the magnitude of muscle responses induced by the SNc and/or RRF stimulation was larger than that induced by the VTA stimulation (Fig. 6*A*; Kruskal–Wallis tests: 100 $\mu$A $\times$ 1, $P = 0.145$; 300 $\mu$A $\times$ 1, $P = 0.180$; 500 $\mu$A $\times$ 1, $P = 8.54 \times 10^{-4}$; 100 $\mu$A $\times$ 3, $P = 0.513$; 300 $\mu$A $\times$ 3, $P = 0.175$; 500 $\mu$A $\times$ 3, $P = 0.00593$; *post hoc* multiple comparisons

with Bonferroni's correction: for 500 $\mu$A $\times$ 1, VTA *vs.* SNc, $P = 8.7585 \times 10^{-4}$; VTA *vs.* RRF, $P = 1.00$; SNc *vs.* RRF, $P = 0.0535$; and for 500 $\mu$A $\times$ 3, VTA *vs.* SNc, $P = 0.0482$; VTA *vs.* RRF, $P = 0.00652$; SNc *vs.* RRF $P = 1$). This result indicates that the SNc and RRF produce stronger muscle responses compared with the VTA and corresponds to the anatomical result showing a larger number of RV-labelled neurons in the SNc and RRF compared with in the VTA (Fig. 2).

Furthermore, to explore the output preference for the proximal *vs.* distal muscles in the VTA, SNc and RRF, we investigated the relationship between the muscle outputs and the stimulus sites. Although there was no clear preference for the proximal (shoulder and elbow) or distal (wrist, digits and intrinsic hand) muscles in the stimulus effect of the VTA, SNc and RRF, the RRF appeared to have divergent outputs to both the proximal and the distal muscles, in comparison to the VTA and SNc (Fig. 6*B*).

### Effect of M1 inactivation on muscle responses induced by VM stimulation

To clarify whether the M1 is involved in the muscle responses evoked by VM stimulation, two monkeys (monkeys D and Y) were subjected to focal inactivation experiments affecting the M1 forelimb region (Table 2) by

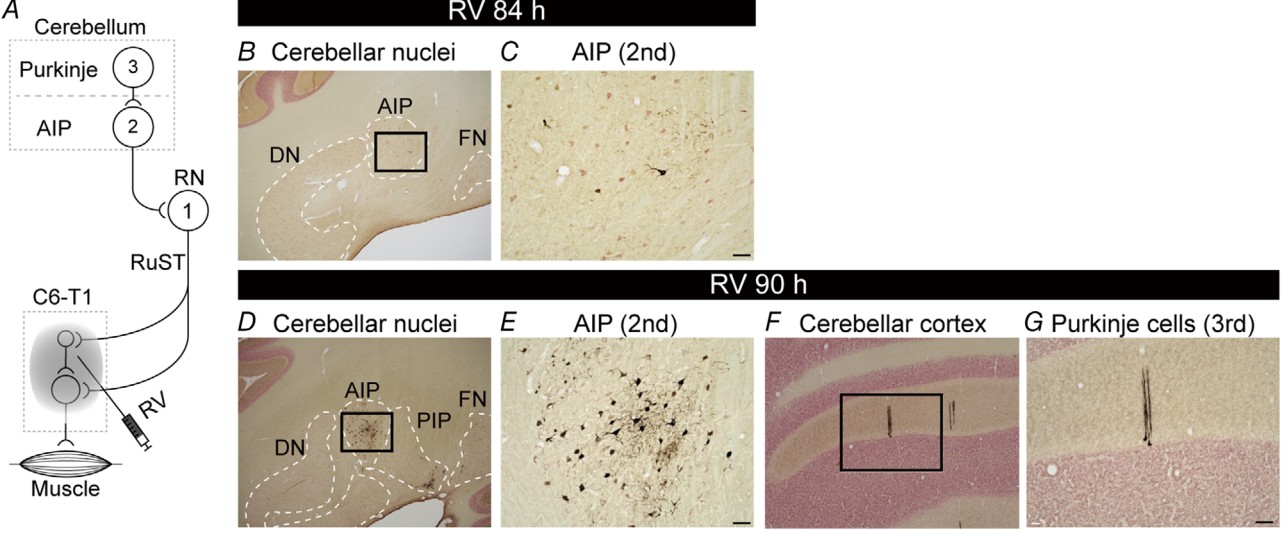

**Figure 3. Retrograde transneuronal labelling from the cervical enlargement in the red nucleus–cerebellar circuit**
*A*, diagram of retrograde trans-synaptic transport of rabies virus from the cervical enlargement. Each number ('1', '2' and '3') indicates the order from the cervical enlargement. *B* and *D*, photomicrographs of the coronal section including the AIP. *C* and *E*, higher magnification photomicrographs of the labelled neurons in AIP. Scale bar, 100 $\mu$m. The magnified areas shown in *C* and *E* correspond to the black boxes in *B* and *D*, respectively. *F*, photomicrograph of the coronal section including cerebellar cortex. *G*, higher magnification photomicrograph of the labelled neurons in Purkinje cells. Scale bar, 100 $\mu$m. The magnified area corresponds to the black box in *F*. Abbreviations: AIP, anterior interpositus nucleus; DN, dentate nucleus; FN, fastigial nucleus; PIP, posterior interpositus nucleus; RN, red nucleus; RuST, rubrospinal tract; RV, rabies virus.

microinjections of muscimol, a GABA$_A$ receptor agonist (5 $\mu$g/$\mu$l, dissolved in 0.1 M phosphate buffer, pH 7.4). Evoked muscle responses were compared before and during M1 inactivation (Fig. 7*A*). To obtain a somatotopic map of the M1 for determination of the inactivation sites, we performed intracortical microstimulation before the inactivation experiments (Fig. 7*B*). Muscimol was injected mainly into wrist and digit regions of the M1, with a total amount of 5–6 $\mu$l (1 $\mu$l per site). The three-pulse stimulation with a current of 300 $\mu$A was delivered to the VM (Fig. 7*C*).

The muscle responses induced by VM stimulation were disrupted during M1 inactivation (Fig. 7*D*). We found that the MPI was substantially decreased during M1 inactivation, in comparison to the pre-inactivation (Fig. 7*F*; Student's paired *t* test, control *vs.* muscimol, $P = 2.08 \times 10^{-9}$). As a control, physiological saline was injected into the same sites as the muscimol. No muscle responses induced by the VM stimulation were diminished after the saline injections (Fig. 7*E*). There was no significant difference in the MPI before and after the

saline injections (Fig. 7*G*; Student's paired *t* test, control *vs.* saline, $P = 0.116$).

These results suggest that the M1 contributes to the VM-driven muscle responses (Fig. 8).

## Discussion

We investigated a neural substrate that might boost spinal motor output driven by the VM. Our results indicate that the VM projects to the spinal cord indirectly and that VM activation modulates activity of the ipsilateral M1, which generates muscle responses in the contralateral forelimb. Furthermore, we found that the M1 contributes to the muscle responses induced by VM activation. The present study provides evidence for the existence of a multisynaptic VM–spinal pathway that might underlie the modulatory control of motor output to forelimb muscles.

We identified multisynaptic (minimally disynaptic) projections anatomically from the VM to the cervical enlargement (Figs 1 and 2). Electrical stimulation of the VM induced muscle responses in the contralateral

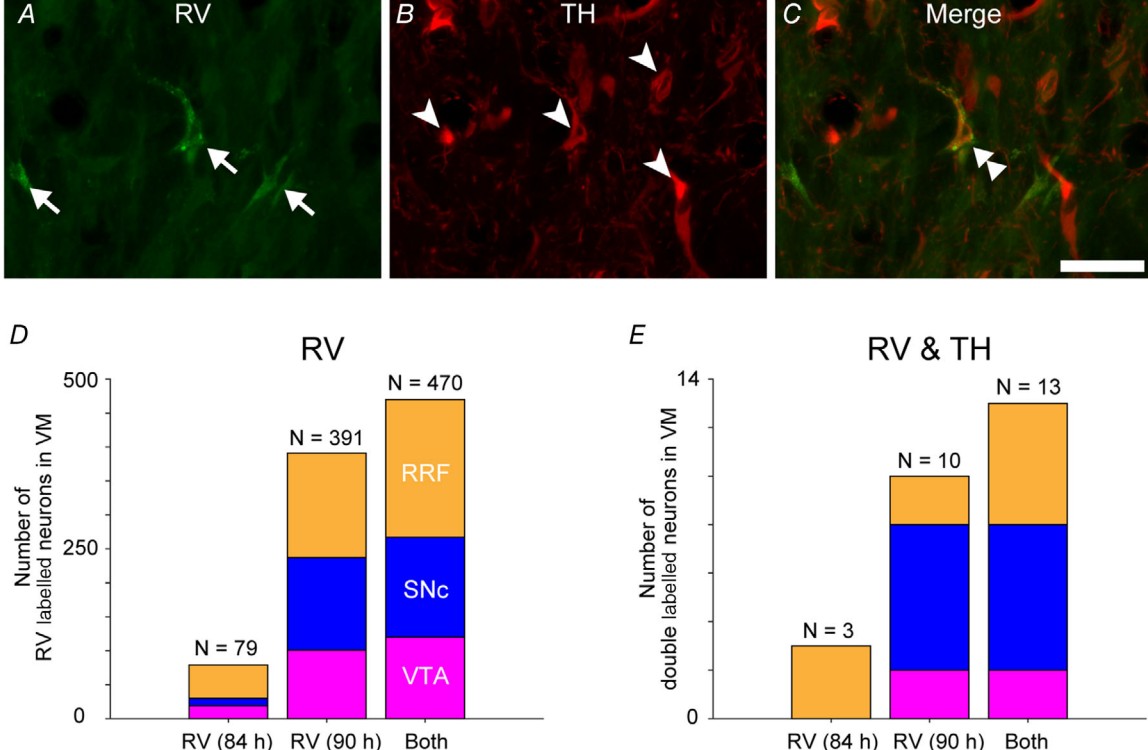

**Figure 4. Double immunofluorescence histochemistry of ventral midbrain neurons for rabies virus and tyrosine hydroxylase**
Representative images of a double-labelled neuron in the RRF. Data were obtained from monkey L (survival time, 90 h). *A*, VM neurons were labelled multisynaptically with RV injected into the cervical enlargement (arrows). *B*, VM neurons immunoreactive for TH (arrowheads). *C*, VM neuron double labelled for RV and TH (merge; double arrowhead). Scale bar, 50 $\mu$m. *D*, distribution of RV-labelled neurons in the VTA, SNc and RRF obtained from immunofluorescence histochemistry. *E*, distribution of double-labelled neurons in the VTA, SNc and RRF obtained from immunofluorescence histochemistry. Abbreviations: RRF, retrorubral field; RV, rabies virus; SNc, substantia nigra pars compacta; TH, tyrosine hydroxylase; VM, ventral midbrain; VTA, ventral tegmental area.

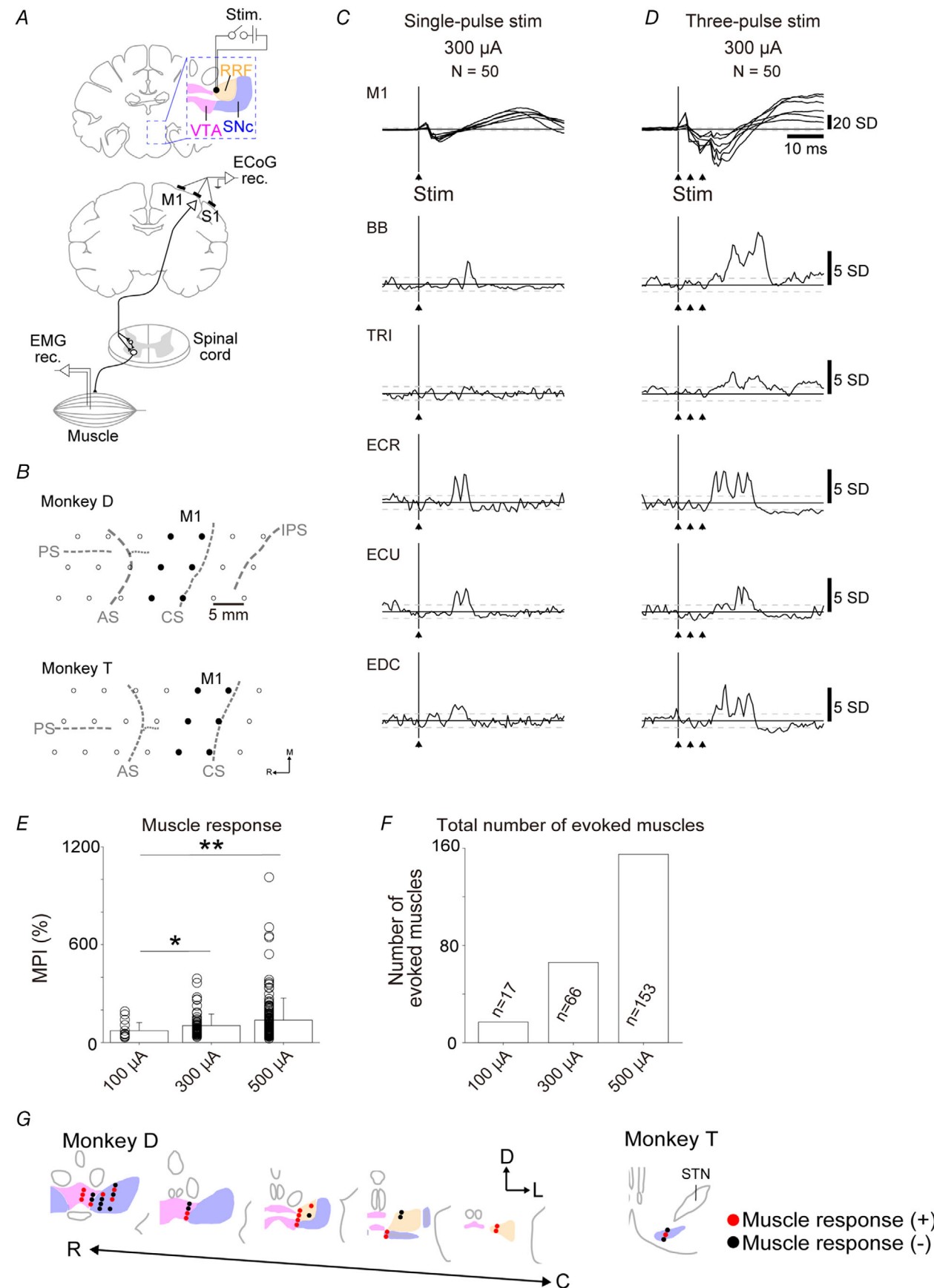

**Figure 5. Responses of the primary motor cortex and muscles evoked by ventral midbrain stimulation**
*A*, schematic diagram of the electrophysiological experiment. The black dot in the RRF indicates the stimulation site for *C* and *D*. *B*, electrode locations for recording cortical responses. Black dots indicate M1 recording sites used for analysis. *C*, representative evoked responses in the M1 and muscles to a single pulse (current intensity, 300 $\mu$A). The number of stimulations for averaging is indicated by 'N'. Arrows indicate the timing of the stimulus pulses. Superimposed waveforms of the recording sites in M1 are shown. Black vertical lines indicate stimulus onset and all signals aligned on this onset. Horizontal dashed lines indicate ±2SD lines based on the baseline activity (from −100 to 0 ms). *D*, representative evoked responses to three-pulse stimulation (current intensity, 300 $\mu$A). Configuration is the same as *C*. Data were obtained from monkey D. *E*, the effect of current intensity with three-pulse stimulation on muscle responses of multiple muscles. The MPI measured the average values between onset and offset of the feature (>2SD) minus the baseline mean and divided by the baseline mean. Data are shown as the mean + SD. Data were obtained from two monkeys (monkeys D and Y). For statistical analysis, the Kruskal–Wallis test (*P* = 0.00210) and multiple comparisons with Bonferroni's *post hoc* test were performed (100 *vs.* 300 $\mu$A, *P* = 2.67 × 10$^{-2}$; 100 *vs.* 500 $\mu$A, *P* = 1.60 × 10$^{-3}$, 300 *vs.* 500 $\mu$A, *P* = 0.716). *$P$ < 0.05 and **$P$ < 0.01. *F*, number of muscles evoked by three-pulse ventral midbrain stimulation. Summation of number of muscles, showing significant evoked responses at each stimulation site. *G*, stimulation sites reconstructed from histological coronal sections in monkeys D and T. Red dots indicate stimulation sites where muscle responses were induced by the three-pulse stimulation with a current of 300 $\mu$A. Black dots indicate stimulation sites where no muscle responses were induced by the three-pulse stimulation with a current of 300 $\mu$A. Abbreviations: AS, arcuate sulcus; BB, biceps brachii; C, caudal; CS, central sulcus; D, dorsal; ECoG, electrocorticogram; ECR, extensor carpi radialis; ECU, extensor carpi ulnaris; EDC, extensor digitorum communis; IPS, intraparietal sulcus; L, lateral; M1, primary motor cortex; MPI, mean percentage increase; PS, principal sulcus; R, rostral; RRF, retrorubral field; S1, primary somatosensory cortex; SNc, substantia nigra pars compacta; STN, subthalamic nucleus; TRI, triceps brachii; VTA, ventral tegmental area.

**Table 3. Number of effective ventral midbrain sites for responses of M1 and forelimb muscles**

| Stimulus | Effective VM sites for M1 response [*n*/total (%)] | Effective VM sites for muscle response [*n*/total (%)] | Number of investigated sites in VM |
|---|---|---|---|
| Current of 100 $\mu$A × 1 | | | |
| Monkey D | 31/31 (100) | 6/31 (19.4) | 31 |
| Monkey T | 5/5 (100) | 0/5 (0) | 5 |
| Current of 100 $\mu$A × 3 | | | |
| Monkey D | 31/31 (100) | 9/31 (29) | 31 |
| Monkey T | 5/5 (100) | 1/5 (20) | 5 |
| Current of 300 $\mu$A × 1 | | | |
| Monkey D | 31/31 (100) | 13/31 (41.9) | 31 |
| Monkey T | 5/5 (100) | 1/5 (20) | 5 |
| Current of 300 $\mu$A × 3 | | | |
| Monkey D | 31/31 (100) | 18/31 (58.1) | 31 |
| Monkey T | 5/5 (100) | 2/5 (40) | 5 |
| Current of 500 $\mu$A × 1 | | | |
| Monkey D | 29/29 (100) | 18/29 (62.1) | 29 |
| Monkey T | 5/5 (100) | 1/5 (20) | 5 |
| Current of 500 $\mu$A × 3 | | | |
| Monkey D | 29/29 (100) | 24/29 (82.8) | 29 |
| Monkey T | 5/5 (100) | 4/5 (80) | 5 |

Abbreviations: M1, primary motor cortex; VM, ventral midbrain.

forelimb (see Fig. 5), indicating that spinal motoneurons are activated by VM stimulation. However, the area that relays the VM to the spinal cord was not known. As part of the mesocortical system projecting to frontal cortical areas, the motor cortex receives inputs from VM neurons in monkeys (Gaspar *et al.* 1992; Williams & Goldman-Rakic, 1998) and rodents (Hosp *et al.* 2011, 2015). In monkeys, the projections to the M1 arise from the caudal VM, predominantly from the RRF and caudal SNc, whereas those to the homologous area in rodents originate from the rostral VM, mainly from the VTA. Our findings of a caudorostral gradient of RV-labelled neurons distributed in the VM (see Fig. 2*B–D*) are consistent with prior data concerning the distribution pattern of VM neurons projecting to the M1 in monkeys. Furthermore, we showed that VM stimulation induces M1 responses (see Fig. 5), which is consistent with an electrophysiological study in rodents (Kunori *et al.* 2014)

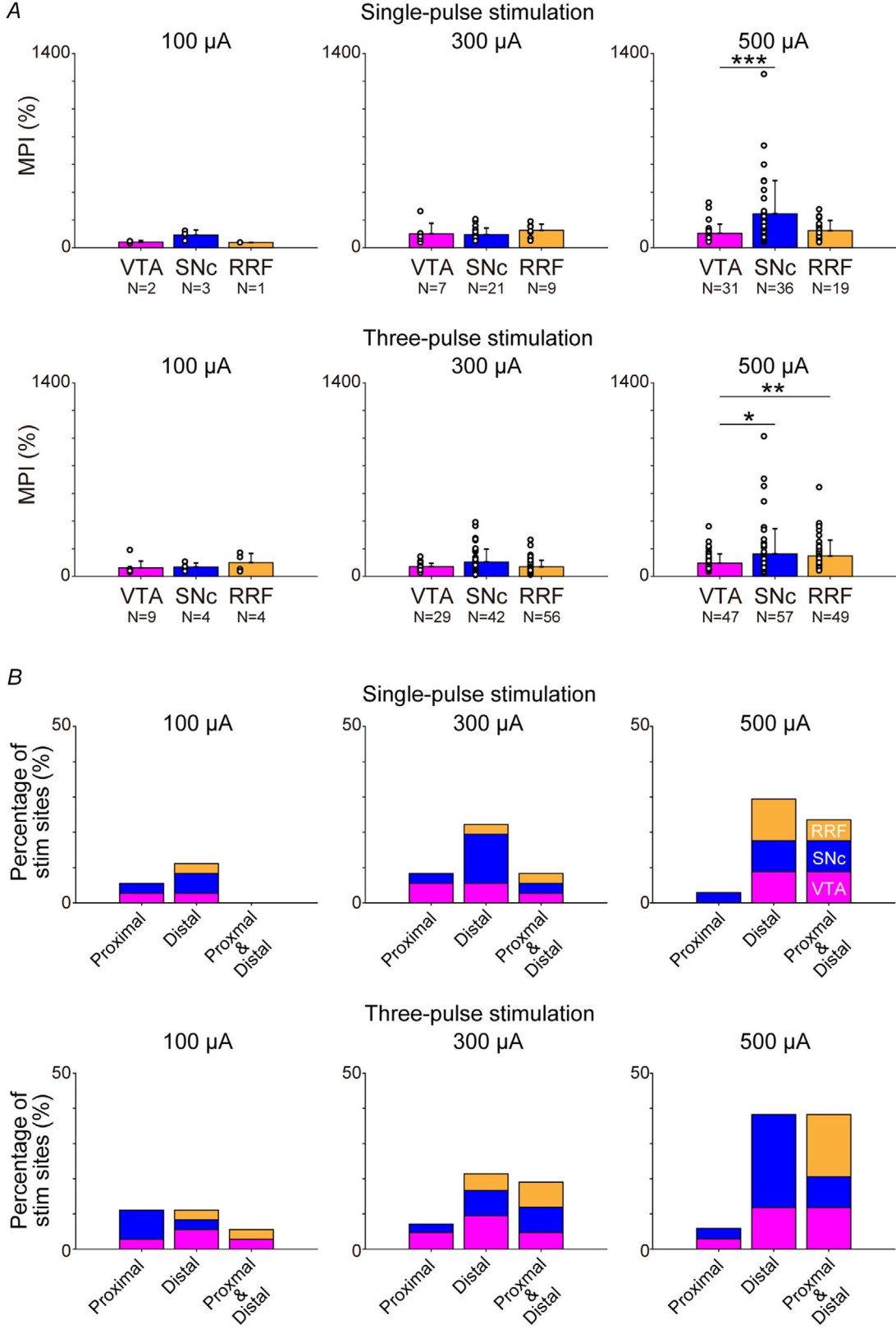

**Figure 6. Regional differences in muscle responses**

*A*, the effect of current intensity and number of pulses on responses of multiple muscles induced by the VTA, SNc and RRF stimulations. Data are shown as the mean + SD. For statistical analysis, Kruskal–Wallis tests were

performed (100 $\mu$A $\times$ 1, $P$ = 0.145; 300 $\mu$A $\times$ 1, $P$ = 0.180; 500 $\mu$A $\times$ 1, $P$ = 8.54 $\times$ 10$^{-4}$; 100 $\mu$A $\times$ 3, $P$ = 0.513; 300 $\mu$A $\times$ 3, $P$ = 0.175; 500 $\mu$A $\times$ 3, $P$ = 0.00593). Multiple comparisons with Bonferroni's *post hoc* test were performed for 500 $\mu$A $\times$ 1 (VTA *vs*. SNc, $P$ = 8.7585 $\times$ 10$^{-4}$; VTA *vs*. RRF, $P$ = 1.00; SNc *vs*. RRF, $P$ = 0.0535) and 500 $\mu$A $\times$ 3 conditions (VTA *vs*. SNc, $P$ = 0.0482; VTA *vs*. RRF, $P$ = 0.00652; SNc *vs*. RRF, $P$ = 1). \**P* < 0.05, \*\**P* < 0.01 and \*\*\**P* < 0.001. *B*, output effects on proximal and distal muscles. The number of stimulus sites that facilitated only proximal muscles (shoulder and elbow), only distal muscles (wrist, digit and intrinsic hand muscles) or a combination of at least one proximal and at least one distal muscle is divided by the total number of stimulation sites (100 $\mu$A $\times$ 1, 100 $\mu$A $\times$ 3 and 300 $\mu$A $\times$ 1, 36 sites; 300 $\mu$A $\times$ 3, 42 sites; 500 $\mu$A $\times$ 1 and 500 $\mu$A $\times$ 3, 34 sites). Data were obtained from three monkeys (monkeys T, D and Y). Given that monkeys Y and D (left hemisphere stimulation) were used for the M1 inactivation study (stimulation parameters, 300 $\mu$A $\times$ 3), data before M1 injection of muscle or saline are included in *A* and *B*. Abbreviations: M1, primary motor cortex; MPI, mean percentage increase; RRF, retrorubral field; SNc, substantia nigra pars compacta; VTA, ventral tegmental area.

and with functional MRI (fMRI) studies in non-human primates (Arsenault *et al*. 2014; Murris *et al*. 2020). Overall, the M1 might be a relay station that mediates the multisynaptic descending pathway from the VM to the cervical enlargement. This is supported by our electrophysiological results showing that M1 inactivation attenuates forelimb muscle responses induced by electrical stimulation of the VM (see Fig. 7).

Other possible candidates that connect the VM to the spinal cord are some descending projections from the brainstem, such as the RN and/or reticular formation. Both the RN (Kennedy *et al*. 1986; Holstege *et al*. 1988; Ralston *et al*. 1988; Burman *et al*. 2000) and the reticular formation (Matsuyama et al, 1997; Davidson *et al*. 2007; Riddle *et al*. 2009; Soteropoulos *et al*. 2012) innervate spinal neurons in the cervical enlargement via the rubrospinal and reticulospinal tract, respectively. It has been reported in rodents that dopaminergic neurons in the VTA and SNc receive inputs from the RN and reticular formation (Watabe-Uchida *et al*. 2012). To the best of our knowledge, however, no reports have described such projections in primates.

In the present study, the onset latencies of muscle responses evoked by the VM stimulation were distributed within the range of 5–25 ms (see Table 4). The minimum latency to activate the forelimb muscles from the M1 is 6 ms, with a median of 8.5–9 ms (see also Park *et al*. 2004). These latencies indicate that the shortest latency to evoke forelimb muscles via VM projection to M1 is 11 ms, considering that the onset of M1 response by VM stimulation is ∼5 ms (see Fig. 5*C* and *D*). Our data also included muscle responses with latencies shorter than 11 ms (see Table 4). Given that the VM is situated in the vicinity of the RN and the cerebral peduncle where the rubrospinal tract originates and the corticospinal tract travels, respectively, these short latencies might indicate an effect of current spread to the rubrospinal and/or corticospinal tracts. Otherwise, activation of unknown VM–brainstem projection(s) leading to the spinal cord might be involved, and such projection(s) can also reflect the RV labelling in the VM. However, our results showed that muscle responses were decreased during M1

inactivation (see Fig. 7). Thus, the M1 might constitute a major relay site that transmits VM signals to spinal motoneurons to generate forelimb muscle activation.

It should also be noted that RV labelling of VM neurons occurred bilaterally (see Fig. 2*B* and *C*). Although the VM projects to the M1 bilaterally (Gaspar *et al*. 1992; Williams & Goldman-Rakic, 1998; Hosp *et al*. 2011, 2015; Zubair *et al*. 2021) and modulates neuronal activity on each side (Arsenault *et al*. 2014; Kunori *et al*. 2014; Murris *et al*. 2020), the ipsilateral projection seems much stronger than the contralateral projection. In the present study, VM stimulation rarely evoked movements from the ipsilateral forelimb (visual inspection, no EMG recording). Considering the ipsilateral predominance of the VM–M1 projection, RV-labelled neurons in the VM contralateral to the spinal injection site must be far greater than on the ipsilateral side. We suggest that ipsilateral labelling in the VM might be attributable to uncrossed and recrossed corticospinal projections (Rosenzweig *et al*. 2009; Yoshino-Saito *et al*. 2010) and/or rather rapid infectious transport of RV from the injection site to the contralateral motoneurons through spinal commissural interneurons (Soteropoulos *et al*. 2013), leading to multisynaptic neuron labelling in the ipsilateral VM. Furthermore, the reticulospinal pathway is known to innervate bilateral spinal neurons (Matsuyama et al, 1997; Davidson *et al*. 2007; Riddle *et al*. 2009; Soteropoulos *et al*. 2012). Thus, if the VM neurons project to the reticulospinal pathway, it is possible to label the VM bilaterally via the reticular formation.

As a technical limitation, in the present study, all cell types, such as dopaminergic, glutamatergic or GABAergic neurons (Yamaguchi *et al*. 2007, 2013), surrounding the electrode tip in the VM can be activated by electrical stimulation. Therefore, the present results cannot be attributed directly to the DA signal alone, as was indicated by a previous study (Murris *et al*. 2020). There is consensus that the mesocortical DA system provides DA innervation over frontal cortical areas (Descarries *et al*. 1987; Lewis *et al*. 1987). Meanwhile, 17% of M1-projecting neurons in the rodent VTA, SNc and RRF are DA neurons (Hosp *et al*. 2015), and approximately

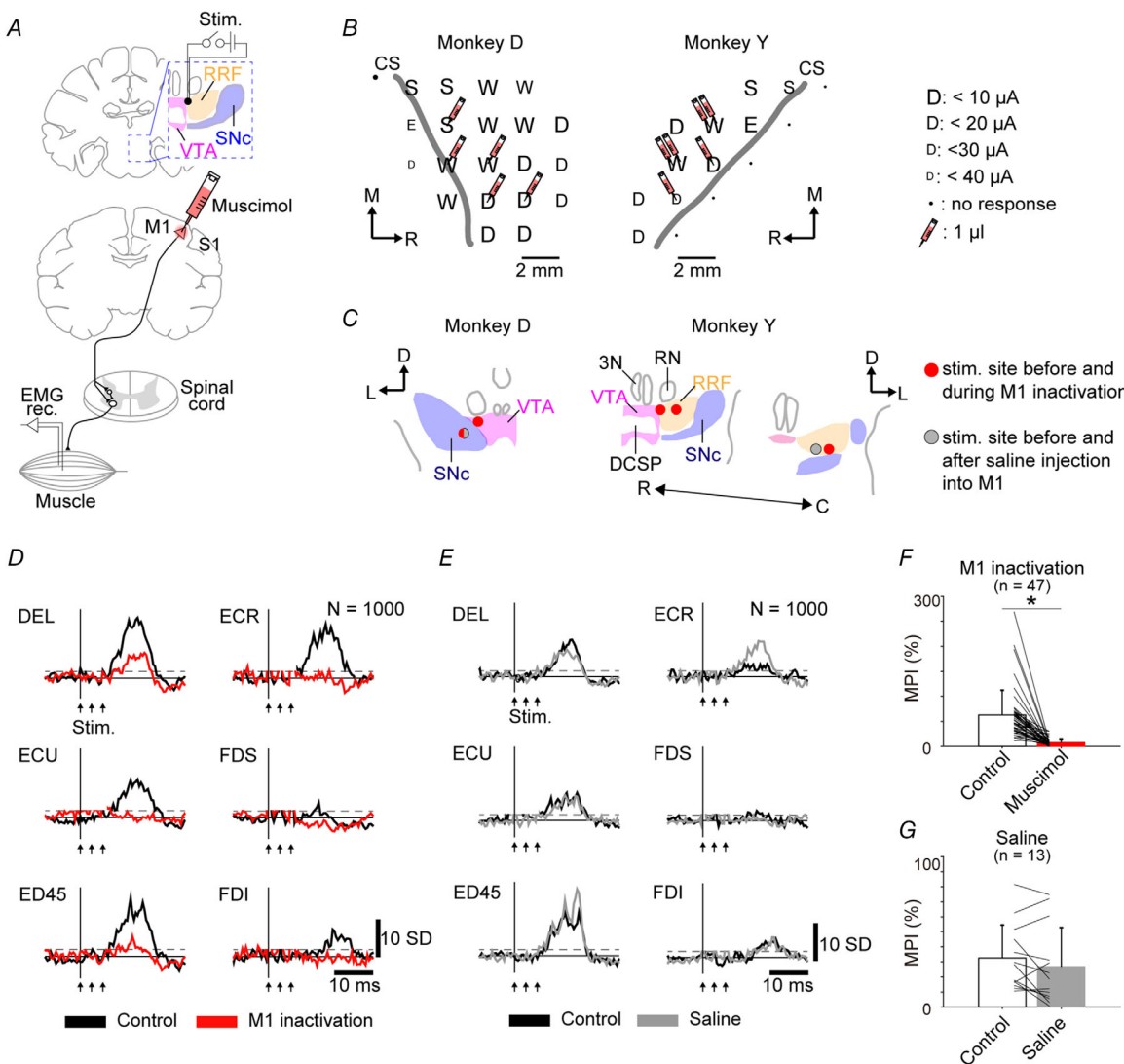

**Figure 7. Effect of primary motor cortex inactivation on muscle responses evoked by ventral midbrain stimulation**

*A*, schematic diagram of M1 inactivation experiment. Muscle responses to ventral midbrain stimulation were investigated before and during M1 inactivation by the microinjection of muscimol. The black dot in the RRF indicates the stimulation site for *D*. *B*, somatotopic maps revealed by intracortical microstimulation and injection sites of muscimol for M1 inactivation. Each electrode penetration is represented with a character indicating the body territory activated at threshold, as follows: D, digit; E, elbow; S, shoulder; and W, wrist. The size of characters indicates the threshold for induction of movements (inset). The sites and volumes of muscimol injection are shown by syringes (1 $\mu$l per syringe). *C*, stimulation sites reconstructed from histological coronal sections in monkeys D and Y. Red dots indicate stimulation sites for M1 inactivation. Grey circles indicate stimulation sites for saline injection into M1. *D*, representative examples of muscle responses obtained from rectified EMGs from the forelimb (three-pulse stimulation with a current of 300 $\mu$A). Black and red traces indicate before (control) and during M1 inactivation, respectively. Stimulus artefacts are hidden in figures. *E*, representative examples of stimulus-triggered averages of muscle responses obtained from rectified EMGs from six forelimb muscles (three-pulse stimulation with a current of 300 $\mu$A). Black and grey traces indicate muscle responses before saline injection into the M1 (control) and after saline injection, respectively. *F*, the effect of M1 inactivation on muscle responses. Data from two monkeys (D and Y) are shown as the mean + SD (Student's paired *t* test for control *vs.* muscimol, *P* = 2.08 × 10$^{-9}$; *\*P* < 0.001). *G*, the effect of the saline injection into the M1 on muscle responses of multiple muscles. Data from two monkeys (D and Y) are shown as the mean + SD (Student's paired *t* test for control *vs.* saline, *P* = 0.116). Abbreviations: C, caudal; D, dorsal; DEL, deltoid; DSCP, decussation of superior cerebellar peduncles; ECR, extensor carpi radialis; ECU, extensor carpi ulnaris; ED45, extensor digitorum 4, 5; EDC, extensor digitorum communis; FDI, first dorsal interosseous; FDS, flexor digitorum superficialis; L, lateral; M1, primary motor cortex; 3N, nucleus of oculomotor nerve; R, rostral; RN, red nucleus; RRF, retrorubral field; S1, primary somatosensory cortex; SNc, substantia nigra pars compacta; VTA, ventral tegmental area.

**Table 4. Latency of muscle responses by ventral midbrain stimulation**

| Joint | Onset latency (ms) | Range of onset latency (ms) | Significant activation [*n*/total (%)] |
|---|---|---|---|
| **Current of 100 $\mu$A × 1** | | | |
| Shoulder | 11.5 | 11.5 | 1/36 (2.7) |
| Elbow | 10.0 | 10.0 | 1/72 (1.4) |
| Wrist | 10.8 ± 3.6 | 7.0–14.0 | 3/180 (1.7) |
| Digit | 13.5 | 13.5 | 1/144 (0.7) |
| Intrinsic | – | – | 0/36 (0) |
| **Current of 100 $\mu$A × 3** | | | |
| Shoulder | 10.8 ± 4.9 | 5.0–17 | 4/36 (11.1) |
| Elbow | 13.3 ± 3.9 | 8.5–18.0 | 4/72 (5.6) |
| Wrist | 11.0 ± 4.9 | 5.5–15.0 | 3/180 (1.7) |
| Digit | 10.4 ± 2.7 | 6.0–13.0 | 6/144 (4.2) |
| Intrinsic | – | – | 0/36 (0) |
| **Current of 300 $\mu$A × 1** | | | |
| Shoulder | 9.8 ± 6.7 | 5.0–14.5 | 2/36 (5.6) |
| Elbow | 13.1 ± 4.8 | 6.5–20.0 | 5/72 (6.9) |
| Wrist | 11.3 ± 2.3 | 7.5–15.5 | 17/180 (9.4) |
| Digit | 11.0 ± 1.6 | 9.0–14.5 | 11/144 (7.6) |
| Intrinsic | 8.0 ± 4.2 | 5.0–11.0 | 2/36 (5.6) |
| **Current of 300 $\mu$A × 3** | | | |
| Shoulder | 16.6 ± 5.5 | 11.0–25.0 | 7/36 (19.4) |
| Elbow | 14.7 ± 3.6 | 8.5–19.0 | 10/72 (13.9) |
| Wrist | 13.0 ± 2.7 | 9.5–19.0 | 23/180 (12.8) |
| Digit | 13.3 ± 4.2 | 6.5–20.5 | 21/144 (14.6) |
| Intrinsic | 18.2 ± 3.2 | 14.0–22.5 | 5/36 (13.9) |
| **Current of 500 $\mu$A × 1** | | | |
| Shoulder | 7.83 ± 2.0 | 6.0–10.0 | 3/34 (8.8) |
| Elbow | 11.3 ± 2.8 | 7.5–17.0 | 11/68 (16.2) |
| Wrist | 10.7 ± 2.6 | 6.5–21.5 | 32/170 (18.8) |
| Digit | 10.2 ± 1.4 | 8.0–14.5 | 29/136 (21.3) |
| Intrinsic | 12.7 ± 1.8 | 9.0–16.0 | 10/34 (29.4) |
| **Current of 500 $\mu$A × 3** | | | |
| Shoulder | 15.0 ± 2.8 | 13.0–17.0 | 2/34 (5.9) |
| Elbow | 13.1 ± 3.2 | 8.5–21.0 | 20/68 (29.4) |
| Wrist | 12.4 ± 3.3 | 9.0–25.0 | 61/170 (35.9) |
| Digit | 12.9 ± 3.0 | 9.0–21.5 | 58/136 (42.6) |
| Intrinsic | 14.8 ± 2.3 | 12.0–19.0 | 12/34 (35.3) |

Values are the mean ± SD. Data were obtained from right forelimb muscles of monkeys D and T. The significant activation column shows the number of muscles exhibiting significant activation in response to ventral midbrain stimulation/total number of joint muscles × stimulation sites investigated (31 or 29 sites in monkey D; 5 stimulation sites in monkey T). Muscles at each joint are follows: shoulder, pectoralis major; elbow, biceps brachii and triceps brachii; wrist, extensor carpi radialis, extensor carpi ulnaris, flexor carpi radialis, flexor carpi ulnaris and palmaris longus; digit, extensor digitorum communis, extensor digitorum 2, 3, flexor digitorum superficialis and flexor digitorum profundus; and intrinsic, first dorsal interosseous.

half of the cortically projecting neurons in the VM are non-dopaminergic in owl monkeys (Gaspar *et al.* 1992). In our work, we failed to detect TH immunoreactivity in many RV-labelled neurons in the VM. Contrary to our expectation, only a limited number of VM neurons were double labelled for both RV and TH (see the Results section) compared with the above previous studies. This observation might result from the RV (Miyachi *et al.* 2006) or adeno-associated virus (Albert *et al.* 2019; Zubair *et al.* 2021) interfering with protein synthesis in infected neurons. It also remains unclear whether slowly acting neurotransmitters/neuromodulators, such as DA, participate in the short-latency cortical responses induced by VM stimulation. The VM contains dopaminergic, glutamatergic and GABAergic neurons (Yamaguchi *et al.* 2007, 2013), which suggests that a proportion of the RV-labelled VM neurons without TH immunoreactivity might be non-dopaminergic. Electrical stimulation studies of the VTA in rodents demonstrated that DA transmission is required for M1 activation (Hosp *et al.* 2011), and other studies suggested that glutamatergic, but not dopaminergic, transmission is required (Kunori *et al.* 2014). Although there is contradiction regarding the neuromodulator of M1 activation, it is intriguing to assume that both DA and glutamate might exert a cooperative action in modulating the activity of corticospinal neurons, as demonstrated by synaptic plasticity in the mesolimbic pathway (Yagishita *et al.* 2014). To elucidate the cell type-specific contribution of a VM-driven modulatory event, cell type-selective optogenetic techniques (Stauffer *et al.* 2016) might be

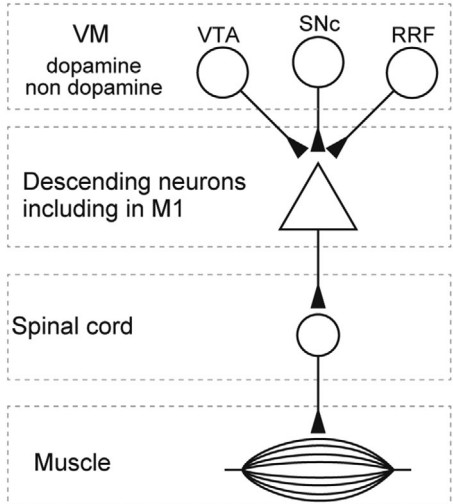

**Figure 8. Proposed multisynaptic pathway linking the ventral midbrain to spinal motoneurons**
The ventral midbrain (VM), consisting of the ventral tegmental area (VTA), substantia nigra pars compacta (SNc) and retrorubral field (RRF), innervates spinal motoneurons multisynaptically via the descending motor pathways.

optimal. Nonetheless, cell type-selective optic stimulation remains difficult in primates (Galvan *et al.* 2017). Thus, electrical stimulation is an alternative technique to investigate the relationship between the VM and spinal motor output.

When comparing our results with those of related previous studies, we found similarities and differences. Kunori *et al.* (2014) also recorded M1 and EMG responses induced by electrical stimulation of the VTA in anaesthetized rats. They found evoked response from the M1, which was consistent with our results, but failed to show activation of forelimb muscles. This might be attributable to their use of single-pulse stimulation at a lower current intensity (maximally 150 $\mu$A). They did not explore the effect of temporal summation on muscle activation systematically. In the present study, we investigated systematically the effect of the number of stimulus pulses and the current intensity. Our results showed that the magnitude and number of evoked muscle responses are associated with the stimulus intensity and number of pulses (see Fig. 5*E* and *F* and Table 3), suggesting that temporal summation might be important in driving the multisynaptic VM–spinal pathway. In relationship to cortical modulation by electrical stimulation in the VTA, activation (bold response) of cortical areas, including the M1, has also been demonstrated in fMRI studies in awake macaques (Arsenault *et al.* 2014; Murris *et al.* 2020), although the cortical responses were decreased in macaques sedated with ketamine (Murris *et al.* 2020). In the present work, in contrast, the short-latency component of local field potentials in the M1 was potentially excitatory, because the M1 local field potentials elicited by the VM stimulation contributed to forelimb muscle activations (see Fig. 7). This discrepancy between our results and fMRI data (Murris *et al.* 2020) might be attributable to differences in recording methods (i.e. electrophysiology *vs.* fMRI). Although fMRI has an excellent spatial resolution, its temporal resolution is inferior to that of electro-physiological techniques. Indeed, the M1 response elicited by the VTA stimulation in sedated conditions represented a brief excitatory response ($\sim$70 ms in duration), followed by a long inhibitory response of $\sim$400 ms (Kunori *et al.* 2014). Thus, the fMRI recordings might not have been able to detect such a short-latency (<100 ms) excitatory response.

In competitive sports, the upregulation of motivation is thought to be crucial for enhancing performance (Tod *et al.* 2003). Positive motivation to obtain reward has been suggested to affect motor performance through the modulation of M1 activity (Marsh *et al.* 2015; Ramakrishnan *et al.* 2017; An *et al.* 2019). Force output depends on the amount of the expected reward (Pessiglione *et al.* 2007). Dopamine (Schultz, 1998; Bromberg-Martin *et al.* 2010) and glutamate (Zell

*et al.* 2020) in the VM are suggested to be sources of motivational signals to obtain reward. A larger amount of reward activates DA neurons more highly (Tobler *et al.* 2005; Takakuwa *et al.* 2017). Our results demonstrates that muscle responses were correlated with stimulus intensity and the number of pulses in VM stimulation (Fig. 5 and Table 3). Furthermore, DA neuron activity is necessary for action initiation (da Silva *et al.* 2018). Thus, the extent of activation of the DA neurons might be important functionally for boosting motor performance. Taken together, we propose that a multisynaptic VM–spinal pathway generates forelimb muscle activity and might constitute a neural substrate underlying the motivational facilitation of motor output. In the context of rehabilitation from motor impairments, motivation to engage in rehabilitation is considered to be crucial for recovery. Animal studies demonstrated that the motivation-related brain area is crucial for motor skill learning (Molina-Luna *et al.* 2009; Hosp *et al.* 2011) and recovery from spinal cord injury (Nishimura *et al.* 2011; Sawada *et al.* 2015; Suzuki et al, 2020). Thus, a multisynaptic VM–spinal pathway might also facilitate (re)learning of motor skills. Moreover, patients with Parkinson's disease resulting from degeneration of DA neurons exhibit deficits in voluntary movements, such as delayed onset and slowness of movements (Jankovic, 2008). The lack of DA input from the VM to descending pathways might explain such motor dysfunctions.

In conclusion, using a combination of anatomical, electrophysiological and pharmacological approaches, we have demonstrated the existence of a multisynaptic VM–spinal pathway via descending pathways that can generate motor output to forelimb muscles. Thus, we propose that the VM is a potential source for boosting descending motor output. Further studies are needed to elucidate the functional roles of the VM–spinal pathway during motor performance, motor learning and rehabilitation from motor impairments and, also, the functional differences among the VTA, SNc and RRF in motivated behaviour and motor control.

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

## Additional information

### Data availability statement

The data that support the findings in this study are included in figures and tables.

### Competing interests

The authors declare no competing financial interests.

### Author contributions

Y.N. and M.S. conceived, designed, performed and analysed the electrophysiological experiments. Y.N., M.S., K-I.I., K.K. and M.T. designed the anatomical experiments. M.S., K-I.I., H.N. and H.I. performed the anatomical experiments. M.S. and K-I.I. analysed the histological data. M.S. and Y.N. prepared figures. M.S., T.I., M.T. and Y.N. wrote the manuscript. All authors have read and approved the final version of this manuscript and agree to be accountable for all aspects of the work in ensuring that questions related to the accuracy or integrity of any part of the work are appropriately investigated and resolved. All persons designated as authors qualify for authorship, and all those who qualify for authorship are listed.

### Funding

This study was supported by the Japan Society for the Promotion of Science to Y.N. (KAKENHI: grant numbers 18H05287, 18H04038 and 15H01818), the Ministry of Education, Culture, Sports, Science and Technology (MEXT) of Japan to Y.N. (a Grant-in-Aid for Scientific Research on Innovative Areas 'Brain Information Dynamics': grant number 18H05151 and 'Hyper Adaptability': grant number 20H05489) and to T.I. (a Grant-in-Aid for Scientific Research on Innovative Areas 'Adaptive Circuit Shift': grant number 26112003), the Japan Science and Technology Agency (Moonshot R&D: grant number JPMJMS2012), the Cooperative Research Program of the Primate Research Institute, Kyoto University (grant numbers H27-A9 and H30-A17) and the Cooperative Study Program (grant number 20-255) of the National Institute for Physiological Sciences to Y.N.

### Acknowledgements

The authors thank N. Takahashi, Y. Yamanishi, T. Kuwahara, K. Isa, S. Hangui, O. Yokoyama and T. Ninomiya for technical support and S. K. Sugawara for comments on the present results.

### Present address

H. Nakagawa: Department of Molecular Neuroscience, World Premier International Immunology Frontier Research Center, Osaka University, Suita, Osaka 565-0871, Japan.

H. Ishida: Schizophrenia Research Project, Tokyo Metropolitan Institute of Medical Science, Setagaya, Tokyo 156-8506, Japan.

### Keywords

motivation, non-human primate, rabies virus, retrorubral field, spinal cord, transneuronal labelling

## Supporting information

Additional supporting information can be found online in the Supporting Information section at the end of the HTML view of the article. Supporting information files available:

**Peer Review History**
**Statistical Summary Document**

