## [Peer Review History · The Journal of Physiology]

A multisynaptic pathway from the ventral midbrain toward spinal motoneurons in monkeys

Michiaki Suzuki, Ken-ichi Inoue, Hiroshi Nakagawa, Hiroaki Ishida, Kenta Kobayashi, Tadashi Isa, Masahiko Takada, and Yukio Nishimura

DOI: 10.1113/JP282429

Corresponding author(s): Yukio Nishimura (nishimura-yk@igakuken.or.jp)

The following individual(s) involved in review of this submission have agreed to reveal their identity: Ken D O'Halloran (Referee #1)

Review Timeline:

Submission Date:	28-Sep-2021
Editorial Decision:	29-Oct-2021
Revision Received:	24-Nov-2021
Editorial Decision:	21-Dec-2021
Revision Received:	23-Dec-2021
Editorial Decision:	05-Jan-2022
Revision Received:	07-Jan-2022
Accepted:	10-Jan-2022

Senior Editor: Richard Carson

Reviewing Editor: Jing-Ning Zhu

Transaction Report:

Dear Dr Nishimura,

Re: JP-RP-2021-282429 "A multisynaptic pathway from the ventral midbrain toward spinal motoneurons in monkeys" by Michiaki Suzuki, Ken-ichi Inoue, Hiroshi Nakagawa, Hiroaki Ishida, Kenta Kobayashi, Tadashi Isa, Masahiko Takada, and Yukio Nishimura

Thank you for submitting your manuscript to The Journal of Physiology. It has been assessed by a Reviewing Editor and by two expert Referees and an Animal Ethics Editor and I am pleased to tell you that it is considered to be acceptable for publication following satisfactory revision.

The reports are copied at the end of this email. Please address all of the points and incorporate all requested revisions, or explain in your Response to Referees why a change has not been made.

NEW POLICY: In order to improve the transparency of its peer review process The Journal of Physiology publishes online as supporting information the peer review history of all articles accepted for publication. Readers will have access to decision letters, including all Editors' comments and referee reports, for each version of the manuscript and any author responses to peer review comments. Referees can decide whether or not they wish to be named on the peer review history document.

Authors are asked to use The Journal's premium BioRender (<https://biorender.com/>) account to create/redrawn their Abstract Figures. Information on how to access The Journal's premium BioRender account is here: <https://physoc.onlinelibrary.wiley.com/journal/14697793/biorender-access> and authors are expected to use this service. This will enable Authors to download high-resolution versions of their figures.

I hope you will find the comments helpful and have no difficulty returning your revisions within 4 weeks.

Your revised manuscript should be submitted online using the links in Author Tasks Link Not Available.

Any image files uploaded with the previous version are retained on the system. Please ensure you replace or remove all files that have been revised.

REVISION CHECKLIST:

- Article file, including any tables and figure legends, must be in an editable format (eg Word)
- Abstract figure file (see above)
- Statistical Summary Document
- Upload each figure as a separate high quality file
- Upload a full Response to Referees, including a response to any Senior and Reviewing Editor Comments;
- Upload a copy of the manuscript with the changes highlighted.

- A potential 'Cover Art' file for consideration as the Issue's cover image;
- Appropriate Supporting Information (Video, audio or data set https://jp.msubmit.net/cgi-bin/main.plex?form_type=display_requirements#supp).

To create your 'Response to Referees' copy all the reports, including any comments from the Senior and Reviewing Editors, into a Word, or similar, file and respond to each point in colour or CAPITALS and upload this when you submit your revision.

I look forward to receiving your revised submission.

If you have any queries please reply to this email and staff will be happy to assist.

Yours sincerely,

REQUIRED ITEMS:

-Author photo and profile. First (or joint first) authors are asked to provide a short biography (no more than 100 words for one author or 150 words in total for joint first authors) and a portrait photograph. These should be uploaded and clearly labelled with the revised version of the manuscript. See Information for Authors for further details.

-You must start the Methods section with a paragraph headed Ethical Approval. A detailed explanation of journal policy and regulations on animal experimentation is given in Principles and standards for reporting animal experiments in The Journal of Physiology and Experimental Physiology by David Grundy J Physiol, 593: 2547-2549. doi:10.1113/JP270818.). A checklist outlining these requirements and detailing the information that must be provided in the paper can be found at: <https://physoc.onlinelibrary.wiley.com/hub/animal-experiments>. Authors should confirm in their Methods section that their experiments were carried out according to the guidelines laid down by their institution's animal welfare committee, and conform to the principles and regulations as described in the Editorial by Grundy (2015). The Methods section must contain details of the anaesthetic regime: anaesthetic used, dose and route of administration and method of killing the experimental animals.

-Your manuscript must include a complete Additional Information section

-Please upload separate high-quality figure files via the submission form.

-A Statistical Summary Document, summarising the statistics presented in the manuscript, is required upon revision. It must be on the Journal's template, which can be downloaded from the link in the Statistical Summary Document section here: https://jp.msubmit.net/cgi-bin/main.plex?form_type=display_requirements#statistics

-Papers must comply with the Statistics Policy https://jp.msubmit.net/cgi-bin/main.plex?form_type=display_requirements#statistics

In summary:

-If n {less than or equal to} 30, all data points must be plotted in the figure in a way that reveals their range and distribution. A bar graph with data points overlaid, a box and whisker plot or a violin plot (preferably with data points included) are acceptable formats.

-If $n > 30$, then the entire raw dataset must be made available either as supporting information, or hosted on a not-for-profit repository e.g. FigShare, with access details provided in the manuscript.

-' n ' clearly defined (e.g. x cells from y slices in z animals) in the Methods. Authors should be mindful of pseudoreplication.

-All relevant ' n ' values must be clearly stated in the main text, figures and tables, and the Statistical Summary Document (required upon revision)

-The most appropriate summary statistic (e.g. mean or median and standard deviation) must be used. Standard Error of the Mean (SEM) alone is not permitted.

-Exact p values must be stated. Authors must not use 'greater than' or 'less than'. Exact p values must be stated to three significant figures even when 'no statistical significance' is claimed.

-Statistics Summary Document completed appropriately upon revision

-A Data Availability Statement is required for all papers reporting original data. This must be in the Additional Information

section of the manuscript itself. It must have the paragraph heading "Data Availability Statement". All data supporting the results in the paper must be either: in the paper itself; uploaded as Supporting Information for Online Publication; or archived in an appropriate public repository. The statement needs to describe the availability or the absence of shared data. Authors must include in their Statement: a link to the repository they have used, or a statement that it is available as Supporting Information; reference the data in the appropriate section(s) of their manuscript; and cite the data they have shared in the References section. Whenever possible the scripts and other artefacts used to generate the analyses presented in the paper should also be publicly archived. If sharing data compromises ethical standards or legal requirements then authors are not expected to share it, but must note this in their Statement. For more information, see our Statistics Policy.

-Please include an Abstract Figure. The Abstract Figure is a piece of artwork designed to give readers an immediate understanding of the research and should summarise the main conclusions. If possible, the image should be easily 'readable' from left to right or top to bottom. It should show the physiological relevance of the manuscript so readers can assess the importance and content of its findings. Abstract Figures should not merely recapitulate other figures in the manuscript. Please try to keep the diagram as simple as possible and without superfluous information that may distract from the main conclusion(s). Abstract Figures must be provided by authors no later than the revised manuscript stage and should be uploaded as a separate file during online submission labelled as File Type 'Abstract Figure'. Please ensure that you include the figure legend in the main article file. All Abstract Figures should be created using BioRender. Authors should use The Journal's premium BioRender account to export high-resolution images. Details on how to use and access the premium account are included as part of this email.

-Author photo and profile. First (or joint first) authors are asked to provide a short biography (no more than 100 words for one author or 150 words in total for joint first authors) and a portrait photograph. These should be uploaded and clearly labelled with the revised version of the manuscript. See Information for Authors for further details.

-

EDITOR COMMENTS

Reviewing Editor:

Ethics Concerns:

Please add more animal experimental details suggested by the Animal Ethics Editor.

Comments to the Author:

Please revise the manuscript according to the reviewers' suggestions, including interpretation of the results and significance, the logic of the manuscript, the presentation of data, the existence of direct VM-spinal projections, the statistic information of labeled neurons, and the animal experimental details regarding to animal Ethics.

Senior Editor:

Comments for Authors to ensure the paper complies with the Statistics Policy:

In revising the manuscript, please pay particularly close attention to all of the guidelines contained in the Statistics Policy for the Journal. For example, the requirement for standard deviations to be reported is to be noted. Furthermore, the authors should consider whether there are opportunities to extend the presentation of raw data. It is necessary also for the statistical summary document must be complete and comprehensive.

I would like to further emphasise the point made by the referees and the Reviewing Editor that certain aspects of the presentation must be tempered or refined, in order to reduce the likelihood that certain aspects of the reported results may be misinterpreted, and ensure that the logic of the investigation (and its interpretation) is made clear. As also noted above, there must be strict adherence to the Statistics Policy for the Journal, and transparent reporting of all relevant data.

REFeree COMMENTS

Referee #1:

Thank you for submitting your manuscript to The Journal of Physiology. Some additional details pertaining to animal ethics and welfare are required.

Please include the source of the animals used in the study.

Line 141: Please include a statement confirming that an adequate plane of anaesthesia was achieved during surgical procedures and how this was determined e.g. absence of pedal withdrawal reflex to stretch and noxious pinch, or cardiorespiratory assessments etc

Line 179: Include the word euthanised or killed in the sentence to confirm death.

Line 247/248. Please describe in full the surgical procedures and post-operative care (if animals were recovered). As above, include a statement in respect of steps taken to ensure adequacy in the depth of anaesthesia during surgical procedures. Although this may appear redundant, the fate of all animals used in the study must be reported in full.

Line 303. Were training sessions used for monkeys studied in the primate chair? Perhaps not, if animals were sedated prior to use. Please clarify in the text.

Line 321: Again, it is necessary to include details of the steps taken to ensure adequacy of depth of sedation during procedures.

Line 378: Include the word euthanised or killed in the sentence to confirm death.

In general, authors are advised that there are no word limits in The Journal and you are advised to report on surgical procedures in full so that it is evident that the highest standards of animal care and welfare were adhered to in the study.

Referee #2:

The manuscript by Suzuki et. al revealed a multisynaptic projection from ventral midbrain (VM) to spinal cord via descending pathway from primary motor cortex (M1), by using retrograde transsynaptic labeling in cervical enlargement. Electric stimulating VM evoked muscle responses, and inactivating M1 neurons diminished muscle responses. Authors have listed a series results identify a VM-spinal pathway mediated by the M1 in monkeys, and they think the pathway should work in motivational control of the spinal motor output. Here are some concerns about current version.

Major comments:

- 1) Motivation to obtain reward could boost motor performance. It is a true and appealing topic. But as the first point highlighted in "Key points summary", the manuscript failed to show strong evidences about the VM-spinal pathway mainly via dopaminergic projection. Even very few data related to this topic in current version. Authors should tune down their tone based on the data.
- 2) Considering a weak linkage between the result in Figure2 and the main topic of this manuscript, it could be more logically to leave Figure2 into supplemental materials.
- 3) Figure1 and 4 provide some example labeled neurons, If authors could add the statistic information of labeled neurons in these figures, like that in Figure3 D, that would help readers to truly understand the manuscript.
- 4) Currently, Figure7 only lists a proposed multisynaptic pathway linking the VM to spinal motoneurons in a very sketchy diagram. More details from this research should be included and presented in the last summarized figure.

Minor comment:

- 1) In Figure6 G, "Muscimol" should be "Saline".

Referee #3:

In their paper, Suzuki et al. investigated anatomical connections between VM and spinal cord. With multi-technical approaches, consisting of employing retrograde transneuronal labeling with the rabies virus and electrical stimulation on VM in anesthetized animals, they were able to propose a neural circuitry able to modulate spinal motor neurons. The manuscript is well organized and methodologically rigorous. However, there seems here and there logical flow of the whole structure might be a bit ambiguous, which could become clearer for readers if the text will be revised incorporating some of the points below into consideration.

1) From the beginning of their manuscript including the abstract, the authors claim that they have demonstrated a multisynaptic pathway that may underlie "motivational control" of the descending motor output. In my opinion in this context and for the experimental procedures used by the authors, it could be more appropriate to use in the whole manuscript terms such as modulatory control of the spinal motor output or modulation of the motor neuron. In other words, motivational control refers to a functional aspect that could be visible during the performance of a particular task. Herein, the electrophysiological experiments were performed with anesthetized monkeys and the use of motivational control is not justified in my opinion.

2) lines 90-91. Again, the authors claim that the overcited findings suggest that motivational signals may affect descending motor pathways to obtain the reward. My feeling is that motivational signals do not affect descending motor pathway "just to obtain reward", but more generally to improve motor performance (e.g. accuracy of the movement, reduction of latencies, increasing of force) modulating, therefore, the motor output.

3) It is not clear in my opinion the rationale underlying the authors' hypothesis. For instance, in the introduction (line 99) they claim that VM can exert a modulatory impact on cortical activities including the M1. Immediately after they claim they want to investigate direct connections between VM and spinal cord. However, it is not clear for the reader this logical jump. Do they want to investigate direct connections between VM (especially VTA) and the spinal cord because there is functional evidence that suggests that this is the case? Or do they hypothesize the presence of direct connections because the red nucleus which is part of the VM has, in turn, direct connections with the spinal cord? Please, the author should better clarify the rationale of their study.

4) With the purpose to demonstrate the anatomical connection between VM and spinal cord, the authors injected into the cervical enlargement respectively BDA and RV tracers. It is clear from Figure 1 that no labeled neurons after BDA injection were found in VTA. However, it is not clear to me, why the authors excluded direct VM-spinal cord connections after RV injection, at all. We should consider some variables that could affect these results. For example, during RV injections we have a higher number of injected sites (6 tracks for BDA and 13 for RV), a higher injection volume (table 1), and, moreover, different injections sites (C6-T1 for RV injections and C6-C7 for BDA injections). Could these variables affect the different results found after BDA and RV injections? This concern could be justified also from electrophysiological results. Indeed, after M1 inactivation it is still possible to observe some weak muscle responses (Figure 6D, DELTOID) during VM stimulation. Could be this reduced muscle response due to direct connections between VM and spinal cord or between VM and premotor cortices?

5) line 486. The authors present data of third-order neurons referring to Purkinje cells. This is an interesting result. However, no mention of its functional role is present in the rest of the manuscript.

6) Figure 6C. The authors show the stimulation sites before and during M1 inactivation. Some sites are in VTA, others in SN while others in RRF. I think that could be interesting to show even in a Supplementary figure the muscles' responses relative to each region. Because of our previous knowledge and also the authors' findings, the prediction is that faster and stronger muscle responses during M1 inactivation should be found when the stimulation site is in RRF when compared with VTA stimulations. This would be more informative for the reader and could improve the authors' results.

END OF COMMENTS

Confidential Review

28-Sep-2021

We greatly appreciate the editors and the three referees for their careful reading of our manuscript and further supportive and helpful comments on our manuscript. We addressed and incorporated all the comments in the revised manuscript. We have made several changes that are marked in blue. Below we list our responses to each of the editors' and referees' comments.

EDITOR COMMENTS

Reviewing Editor:

Ethics Concerns:

Please add more animal experimental details suggested by the Animal Ethics Editor.

Response

We added detail descriptions about animal experiments suggested by the Animal Ethics Editor (Referee#1).

Comments to the Author:

Please revise the manuscript according to the reviewers' suggestions, including interpretation of the results and significance, the logic of the manuscript, the presentation of data, the existence of direct VM-spinal projections, the statistic information of labeled neurons, and the animal experimental details regarding to animal Ethics.

Response

We responded to the Referees' comments and modified the text according to their suggestions.

Senior Editor:

Comments for Authors to ensure the paper complies with the Statistics Policy: In revising the manuscript, please pay particularly close attention to all of the guidelines contained in the Statistics Policy for the Journal. For example, the requirement for standard deviations to be reported is to be noted. Furthermore,

the authors should consider whether there are opportunities to extend the presentation of raw data. It is necessary also for the statistical summary document must be complete and comprehensive.

Response

P values are stated to three significant figures in the main text and figure legend (Figure 5, 6 and 7) and standard deviations are also stated in Figures 5, 6, 7 and Table3. We completed the statistical summary report file (submitted with the revised manuscript). All raw data are presented in the Figures and Tables. Corrections are listed as below.

Page 51-53, Line 1049-1075 (Figure legends)

Figure 5. Responses of the M1 and muscles evoked by VM stimulation. ... Data are shown as the mean + SD. Data were obtained from two monkeys (Monkey D and Y).

For statistical analysis, Kruskal-Wallis test ($P = 0.00210$) and post hoc multiple comparisons with Bonferroni's were performed ($P_{100\mu A}$ vs $P_{300\mu A} = 2.67 \times 10^{-2}$, $P_{100\mu A}$ vs $P_{500\mu A} = 1.60 \times 10^{-3}$, $P_{300\mu A}$ vs $P_{500\mu A} = 0.716$). *, $P < 0.05$, **, $P < 0.01$

Page 53-54, Line 1095-1120 (Figure legends)

Figure 7. Effect of M1 inactivation on muscle responses evoked by VM stimulation.

... (F) The effect of M1 inactivation on muscle responses. **Data are shown as the mean + SD.** Data obtained from two monkeys (Monkey D and Y). For statistical analysis, paired t-test was performed (P_{control} vs $P_{\text{muscimol}} = 2.08 \times 10^{-9}$, *, $P < 0.001$). (G) The effect of the saline injection into the M1 on muscle responses of multiple muscles. **Data are shown as the mean + SD.** Data obtained from two monkeys (Monkey D and Y). For statistical analysis, paired t-test was performed (P_{control} vs $P_{\text{saline}} = 0.116$). ...

Table 3. Latency of muscle responses by VM stimulation.

Current of 100 $\mu\text{A} \times 1$			
Joint	Onset latency (ms)	Range of onset latency (ms)	N
Shoulder	11.5	11.5	1/36 (2.7%)
Elbow	10.0	10.0	1/72 (1.4%)
Wrist	10.8 \pm 3.6	7.0 - 14.0	3/180 (1.7%)
Digit	13.5	13.5	1/144 (0.7%)
Intrinsic	—	—	0/36 (0%)
Current of 100 $\mu\text{A} \times 3$			
Shoulder	10.8 \pm 4.9	5.0 - 17	4/36 (11.1%)
Elbow	13.3 \pm 3.9	8.5 - 18.0	4/72 (5.6%)
Wrist	11.0 \pm 4.9	5.5 - 15.0	3/180 (1.7%)
Digit	10.4 \pm 2.7	6.0 - 13.0	6/144 (4.2%)
Intrinsic	—	—	0/36 (0%)
Current of 300 $\mu\text{A} \times 1$			
Shoulder	9.8 \pm 6.7	5.0 - 14.5	2/36 (5.6%)
Elbow	13.1 \pm 4.8	6.5 - 20.0	5/72 (6.9%)
Wrist	11.3 \pm 2.3	7.5 - 15.5	17/180 (9.4%)
Digit	11.0 \pm 1.6	9.0 - 14.5	11/144 (7.6%)
Intrinsic	8.0 \pm 4.2	5.0 - 11.0	2/36 (5.6%)
Current of 300 $\mu\text{A} \times 3$			
Shoulder	16.6 \pm 5.5	11.0 - 25.0	7/36 (19.4%)
Elbow	14.7 \pm 3.6	8.5 - 19.0	10/72 (13.9%)
Wrist	13.0 \pm 2.7	9.5 - 19.0	23/180 (12.8%)
Digit	13.3 \pm 4.2	6.5 - 20.5	21/144 (14.6%)
Intrinsic	18.2 \pm 3.2	14.0 - 22.5	5/36 (13.9%)
Current of 500 $\mu\text{A} \times 1$			
Shoulder	7.83 \pm 2.0	6.0 - 10.0	3/34(8.8%)
Elbow	11.3 \pm 2.8	7.5 - 17.0	11/68 (16.2%)
Wrist	10.7 \pm 2.6	6.5 - 21.5	32/170 (18.8%)
Digit	10.2 \pm 1.4	8.0 - 14.5	29/136 (21.3%)
Intrinsic	12.7 \pm 1.8	9.0 - 16.0	10/34 (29.4%)
Current of 500 $\mu\text{A} \times 3$			
Shoulder	15.0 \pm 2.8	13.0 - 17.0	2/34(5.9%)
Elbow	13.1 \pm 3.2	8.5 - 21.0	20/68 (29.4%)
Wrist	12.4 \pm 3.3	9.0 - 25.0	61/170 (35.9%)
Digit	12.9 \pm 3.0	9.0 - 21.5	58/136 (42.6%)
Intrinsic	14.8 \pm 2.3	12.0 - 19.0	12/34 (35.3%)

Values are mean \pm standard deviation (SD). ...

I would like to further emphasise the point made by the referees and the Reviewing Editor that certain aspects of the presentation must be tempered or

refined, in order to reduce the likelihood that certain aspects of the reported results may be misinterpreted, and ensure that the logic of the investigation (and its interpretation) is made clear. As also noted above, there must be strict adherence to the Statistics Policy for the Journal, and transparent reporting of all relevant data.

Response

We responded to the Referees' comments and modified the text in order to make the logic of the manuscript clear. Following the statistical policy for the journal we transparently reported all relevant data in the manuscript and the statistical reporting summary file.

REFEREE COMMENTS

Referee #1:

Thank you for submitting your manuscript to The Journal of Physiology. Some additional details pertaining to animal ethics and welfare are required.

1-1) Please include the source of the animals used in the study.

Response

We added an additional description about the source of the animals in Animal and Ethical approval section in Methods as follow.

Page 8, Line 128-130 (Methods)

Six macaque monkeys (5.0-8.0 kg) were used for the present study (Table 1). All monkeys were supplied by National BioResource Project or Kawahara Bird-Animal Trading Co Ltd. We performed neuroanatomical ...

1-2) Line 141: Please include a statement confirming that an adequate plane of anaesthesia was achieved during surgical procedures and how this was determined e.g. absence of pedal withdrawal reflex to stretch and noxious pinch, or cardiorespiratory assessments etc

Response

We confirmed the depth of anesthesia by pain response, and monitored vital signs during surgery. We added additional descriptions about the referee's points as follows.

Page 9, Line 165-188

Surgery for viral injections into the spinal cord. The surgeries described below were performed under general anesthesia initiated by ketamine (10 mg/kg, i.m.) plus xylazine (1 mg/kg, i.m.) and maintained with sodium pentobarbital (20 mg/kg, i.v.) or 1-1.5% isoflurane. The depth of anesthesia was confirmed by pain response. During anesthesia, the animal's vital signs, such as respiratory/circulatory parameters and body temperature, were carefully monitored. All surgical procedures were performed in sterile conditions. The skin and axial muscles were dissected at the level of the C3 to T2 vertebrae. ... After all injections were completed, muscle and skin were incised.

Postoperative management consisted of observation of the operated animals until they were completely recovered from the anesthesia, and administration of dexamethasone (0.825 mg/kg, i.m.), ampicillin (40 mg/kg, i.m.), and ketoprofen (2.0 mg/kg, i.m.). Daily observation of the animals for evaluation of their health status was conducted by veterinarians as well as by researchers. Survival time after virus injection was set...

1-3) Line 179: Include the word euthanised or killed in the sentence to confirm death.

Response

We added additional descriptions about euthanasia and death definition in Histological procedure section in neuroanatomical experiments in Methods.

Page 10, Line 190-197

Histological procedures. After the survival periods after viral injections, the monkeys were deeply anesthetized with an overdose of sodium pentobarbital (50 mg/kg, i.v.). After confirmation of loss of pain reflex under deep anesthesia, the monkeys were euthanized by transcatheter perfusion with 0.1 M phosphate-buffered saline (PBS), followed by 10% formalin dissolved in 0.1 M PB. Then, we confirmed the death of animals by confirming respiratory arrest, cardiac arrest, and pupillary dilatation. The fixed brain and spinal cord were removed from the skull, postfixed in the same fresh fixative overnight at 4°C, and equilibrated with 30% sucrose in 0.1 M PB at 4°C. ...

1-4) Line 247/248. Please describe in full the surgical procedures and post-operative care (if animals were recovered). As above, include a statement in respect of steps taken to ensure adequacy in the depth of anaesthesia during surgical procedures. Although this may appear redundant, the fate of all animals used in the study must be reported in full.

Response

As described above, we added additional descriptions about surgical procedures including a statement in respect of steps taken to ensure adequacy in the depth of anaesthesia during surgical procedures and post-operative care.

Page 13-14, Line 269-342

Surgery for chronic implants. The surgeries described below were performed under general anesthesia initiated by ketamine (10 mg/kg, i.m.) plus xylazine (1 mg/kg, i.m.) and maintained with 1-1.5% isoflurane. The depth of anesthesia was confirmed by pain response. During anesthesia, the animal's vital signs, such as respiratory/circulatory parameters and body temperature, were carefully monitored. All surgical procedures were performed in sterile conditions.

Implant of a chamber for VM stimulation: ... Scans were carried out under deep anesthesia, which was introduced by ketamine (10 mg/kg, i.m.) plus xylazine (1 mg/kg, i.m.) and maintained with sodium pentobarbital (20 mg/kg, i.m.). The depth of anesthesia was confirmed by pain response. T1-weighted images were collected ... The chamber and titanium tubes were fixed to the screws with acrylic resin. Postoperative management consisted of observation of the operated animals until they were completely recovered from the anesthesia, and administration of ampicillin (40 mg/kg, i.m.), and ketoprofen (2.0 mg/kg, i.m.). Daily observation of the animals for evaluation of their health status was conducted by veterinarians as well as by researchers.

Implant of microwires for EMG recordings: ... The implanted muscles were identified by stimulating through each electrode pair and observing the evoked movements during the surgery. Postoperative management consisted of observation of the operated animal until it was completely recovered from anesthesia, administration of, ampicillin (40 mg/kg, i.m.), and ketoprofen (2.0 mg/kg, i.m.). Daily observation of the animals for evaluation of their health status were conducted by veterinarians and researchers.

Implant of ECoG arrays for cortical activity recordings: ... After implanting the array, the opening of the skull was covered, and a connector was fixed to the skull with acrylic resin. Postoperative management consisted of observation of the operated animal until it was completely recovered from anesthesia, administration of dexamethasone (0.825 mg/kg, i.m.), ampicillin (40 mg/kg, i.m.), and ketoprofen (2.0 mg/kg, i.m.). Daily observation of the animals for evaluation of their health status were conducted by veterinarians and researchers.

Implant of a chamber for M1 inactivation: ... fixed to the skull with acrylic resin. Postoperative management consisted of observation of the operated animal until it was completely recovered from anesthesia, administration of ampicillin (40 mg/kg, i.m.), and ketoprofen (2.0 mg/kg, i.m.). Daily observation of the animals for evaluation of their health status were conducted by veterinarians and researchers.

1-5) Line 303. Were training sessions used for monkeys studied in the primate

chair? Perhaps not, if animals were sedated prior to use. Please clarify in the text.

Response

We trained animals to sit in a monkey chair. We added descriptions as follows in Electrical stimulation to the VM section in Methods.

Page 16, Line 339-342

Electrical stimulation to the VM. Prior to data collection, the monkeys were trained to sit in a primate chair. After completion of chair training, experiments were conducted. The monkeys were sedated with ketamine (10 mg/kg, i.m.), and their heads were fixed to the primate chair. The depth of anesthesia was confirmed by pain response. ...

1-6) Line 321: Again, it is necessary to include details of the steps taken to ensure adequacy of depth of sedation during procedures.

Response

We trained animals to sit in a monkey chair. We added descriptions as follows in M1 inactivation section in Methods.

Page 17, Line 362

M1 inactivation. To determine the injection sites in Monkey D and Y, somatotopic maps were investigated using intracortical microstimulation (ICMS) under sedation with ketamine (10 mg/kg, i.m.) as described in our previous study (Nishimura et al., 2007). The depth of anesthesia was confirmed by pain response. A tungsten microelectrode (1.1-1.5 M Ω at 1 kHz) was inserted perpendicularly to the cortical surface using a hydraulic micromanipulator. ...

1-7) Line 378: Include the word euthanised or killed in the sentence to confirm death.

Response

We added additional descriptions about euthanasia and death definition in Histological procedure section in electrophysiological experiments in Methods.

Page 19-20, Line 414-423

Histological confirmation of stimulation sites. At the end of the experiment on each animal, the monkeys were deeply anesthetized with an overdose of sodium pentobarbital (50 mg/kg, i.v.) to make electrocoagulation for identification of stimulation sites in the VM. After confirmation of loss of pain reflex under deep anesthesia, the electrocoagulation was made using rectangular constant current at 30 μ A for 20 s through the stimulating electrode. Then, the monkeys were euthanized by transcardial perfusion with 0.1 M PBS, followed by 10% formalin dissolved in 0.1 M PB. Then, we confirmed the death of animals by confirming respiratory arrest, cardiac arrest, and pupillary dilatation. The perfused brain was removed ...

1-8) In general, authors are advised that there are no word limits in The Journal and you are advised to report on surgical procedures in full so that it is evident that the highest standards of animal care and welfare were adhered to in the study.

Response

As Referee#1's suggestions. we added additional descriptions about surgical procedures, euthanasia and death definition in Methods.

Referee #2:

The manuscript by Suzuki et. al revealed a multisynaptic projection from ventral midbrain (VM) to spinal cord via descending pathway from primary motor cortex (M1), by using retrograde transsynaptic labeling in cervical enlargement. Electric stimulating VM evoked muscle responses, and inactivating M1 neurons diminished muscle responses. Authors have listed a series results identify a VM-spinal pathway mediated by the M1 in monkeys, and they think the pathway should work in motivational control of the spinal motor output. Here are some concerns about current version.

Major comments:

1) Motivation to obtain reward could boost motor performance. It is a true and appealing topic. But as the first point highlighted in "Key points summary", the manuscript failed to show strong evidences about the VM-spinal pathway mainly via dopaminergic projection. Even very few data related to this topic in current version. Authors should tune down their tone based on the data.

Response

The number of double labeled neurons by anti-RV and anti-TH are extremely small compared to previous studies examining the percentage of dopaminergic neurons projecting to M1 using conventional neural tracer (Gaspar et al., 1992). This may be because the RV interferes with protein synthesis in infected neurons (Miyachi et al., 2006). Thus, our results probably underestimate dopaminergic neurons projecting mutisynaptically to the spinal cord. Therefore, what we can state here is the existence of dopaminergic neurons which multisynaptically project to the spinal cord.

Second, as a technical limitation, all cell types, such as dopaminergic, glutamatergic, or GABAergic neurons (Yamaguchi et al., 2007; 2013), surrounding the electrode tip in the VM can be activated by electrical stimulation. To elucidate the dopamine specific contribution of a VM-driven modulatory event, cell-type-selective optogenetic techniques (Stauffer et al., 2016) may be optimal as discussed in Discussion.

Finally, the recent rodent study demonstrated that VTA glutamate neuron activity serves

as a reinforcer independent from dopamine to control reward-seeking behavior (Zell et al., 2020). Therefore, not only dopamine but also glutamate neurons is involved in motivational processing.

Considering the above and Referee#2's and Referee#3's comments (1), we refrained from using the terms "dopamine" and "motivational control" too much, and now modified the text to tone down regarding dopamine.

Page 3, Line 54-55 (Key points summary)

Motivation to obtain reward is thought to boost motor performance, and the activity in the ventral midbrain is important to the motivational process.

Page 3, Line 62-63 (Key points summary)

A multisynaptic VM–spinal pathway most likely plays a pivotal role in modulatory control of the spinal motor output.

Page 5, Line 86 (Abstract)

... Thus, the present study has identified a multisynaptic VM–spinal pathway that, at least in part, is mediated by the M1 and may play a pivotal role in modulatory control of the spinal motor output.

Page 6, Line 102-106 (Introduction)

... The activity of the ventral tegmental area (VTA) and the substantia nigra pars compacta (SNc), which predominantly contain dopaminergic cells, plays an important role in processing motivation (Schultz, 1998; Bromberg-Martin et al., 2010). Not only dopaminergic but also glutamatergic neurons serve as a reinforcer independent from dopamine to control reward-seeking behavior (Zell et al., 2020). ...

Page 7, Line 125 (Introduction)

... Our overall results have demonstrated a multisynaptic VM–spinal pathway that may underlie modulatory control of the descending motor output to forelimb muscles.

Page 28, Line 592-593(Discussion)

The present study provides evidence for the existence of a multisynaptic VM–spinal pathway that may underlie the modulatory control of motor output toward forelimb muscles.

Page 33, Line 723-726 (Discussion)

... Taken together, **we propose** that a multisynaptic VM–spinal pathway generates forelimb muscle activity might constitute a neural substrate underlying the motivational facilitation of motor output. ...

2) Considering a weak linkage between the result in Figure2 and the main topic of this manuscript, it could be more logically to leave Figure2 into supplemental materials.

Response: As Referee#2 commented, we would like to arrange Figure2 into supplementary materials. However, a following sentence is described in information for Authors “The Journal does not usually accept supplemental/supporting material that cannot be included with the body of the manuscript.” Thus, we embed Figure2 into the main manuscript.

3) Figure1 and 4 provide some example labeled neurons, If authors could add the statistic information of labeled neurons in these figures, like that in Figure3 D, that would help readers to truly understand the manuscript.

Response

Thank you for such constructive suggestion. We made the new graph showing the number of RV-labeled neurons obtained from immunofluorescence histochemistry with anti-RV antibody (Figure 4D) and that of double (anti-RV and anti-TH) labeled neurons in each region (Figure 4E). Additional information regarding the number of labeled neuron each area (VTA, SNc, RRF) also was described in the manuscript. As we described in Discussion, it should be noted again that only a limited number of VM neurons were double-labeled for both RV and TH. This observation may result from the fact that RV interferes with the protein synthesis in infected neurons (Miyachi et al., 2006).

Page 24, Line 512-519

When double immunofluorescence histochemistry for RV and TH was performed, we found that only a small population of RV-labeled neurons displayed TH

immunoreactivity [3.8% (3/79 neurons) in Monkey F and 2.6% (10/391) in Monkey L] (Fig. 4): for Monkey F, 0% (0/19) in the VTA, 0% (0/11) in the SNc, and 6.1% (3/49) in the RRF; and for Monkey L, 2.0% (2/101) in the VTA, 4.4% (6/136) in the SNc, and 1.3% (2/154) in the RRF. Since RV interferes with protein synthesis in infected neurons (Miyachi et al., 2006), it should be noted that this result probably underestimates the number of double-labeled neurons.

Page 51, Line 1039-1047 (Figure legends)

Figure 4. Double immunofluorescence histochemistry of VM neurons for RV and TH. Representative images of a double-labeled neuron in the RRF. (A) VM neurons were labeled multisynaptically with RV injected into the cervical enlargement (RV; arrows). (B) VM neurons immunoreactive for TH (TH; arrowheads). (C) VM neuron double-labeled for RV and TH (Merge; double arrow). Data obtained from Monkey L (survival time, 90 h). Scale bar, 50 μ m. (D) Distribution of RV-labeled neurons in the

VTA, SNc, and RRF obtained from immunofluorescence histochemistry. (E)

Distribution of double labeled neurons in the VTA, SNc, and RRF obtained from immunofluorescence histochemistry.

4) Currently, Figure7 only lists a proposed multisynaptic pathway linking the VM to spinal motoneurons in a very sketchy diagram. More details from this research should be included and presented in the last summarized figure.

Response: Thank you for suggestion. We modified Figure 7 as follow. The figure described multisynaptic connection from VTA, SNc, RRF including non-dopamine/dopamine neurons in the VM section to the spinal cord. Since we added new Figure as Figure 7 to response Referee#3's comment, original Figure 7 has moved to Figure 8.

Page 54-55, Line 1122-1124 (Figure legends)

Figure 8. Proposed multisynaptic pathway linking the VM to spinal motoneurons. The VM consisting of the VTA, SNc, and RRF multisynaptically innervates spinal motoneurons via the descending motor pathways.

Minor comment:

1) In Figure6 G, "Muscimol" should be "Saline".

Response

Thank you for noticing that. We corrected it and added SD error bars in Figure 7G. To answer Referee#3's comment, a new Figure has been added as Figure 6, so the original Figure 6 has been changed to Figure 7.

Page 53-54, Line 1095-1120 (Figure legends)

Figure 7. Effect of M1 inactivation on muscle responses evoked by VM stimulation. ... (F) The effect of M1 inactivation on muscle responses. Data are shown as the mean + SD. Data obtained from two monkeys (Monkey D and Y). For statistical analysis, paired t-test was performed (P_{control} vs $P_{\text{muscimol}} = 2.08 \times 10^{-9}$, *; $P < 0.001$). (G) The effect of the saline injection into the M1 on muscle responses of multiple muscles. Data are shown as the mean + SD. Data obtained from two monkeys (Monkey D and Y). For

statistical analysis, paired t-test was performed (P_{control} vs $P_{\text{saline}} = 0.116$).
Abbreviations: C, caudal; D, dorsal; L, lateral; R, rostral; 3N, nucleus of oculomotor nerve.

Referee #3:

In their paper, Suzuki et al. investigated anatomical connections between VM and spinal cord. With multi-technical approaches, consisting of employing retrograde transneuronal labeling with the rabies virus and electrical stimulation on VM in anesthetized animals, they were able to propose a neural circuitry able to modulate spinal motor neurons. The manuscript is well organized and methodologically rigorous. However, there seems here and there logical flow of the whole structure might be a bit ambiguous, which could become clearer for readers if the text will be revised incorporating some of the points below into consideration.

1) From the beginning of their manuscript including the abstract, the authors claim that they have demonstrated a multisynaptic pathway that may underlie "motivational control" of the descending motor output. In my opinion in this context and for the experimental procedures used by the authors, it could be more appropriate to use in the whole manuscript terms such as modulatory control of the spinal motor output or modulation of the motor neuron. In other words, motivational control refers to a functional aspect that could be visible during the performance of a particular task. Herein, the electrophysiological experiments were performed with anesthetized monkeys and the use of motivational control is not justified in my opinion.

Response

Thank you for such constructive criticism. The electrophysiological experiments in the present study were conducted in anesthetized animals. We agree that experiments in behaving monkeys must be necessary to demonstrate the motivational control of motor outputs. The term "motivational control" is our speculation, based on previous studies that the ventral midbrain is important in processing motivation. Regarding the term "motivational control", we now modified the text to "modulatory control of the spinal motor output".

Page 3, Line 62-63 (Key points summary)

A multisynaptic VM–spinal pathway most likely plays a pivotal role in modulatory control of the spinal motor output.

Page 5, Line 84-86 (Abstract)

... Thus, the present study has identified a multisynaptic VM–spinal pathway that, at least in part, is mediated by the M1 and may play a pivotal role in modulatory control of the spinal motor output.

Page 7, Line 123-125 (Introduction)

... Our overall results have demonstrated a multisynaptic VM–spinal pathway that may underlie modulatory control of the descending motor output to forelimb muscles.

Page 28, Line 591-593 (Discussion)

... The present study provides evidence for the existence of a multisynaptic VM–spinal pathway that may underlie the modulatory control of motor output toward forelimb muscles.

Page 33, Line 723-726 (Discussion)

... Taken together, we propose that a multisynaptic VM–spinal pathway generates forelimb muscle activity might constitute a neural substrate underlying the motivational facilitation of motor output. ...

2) lines 90-91. Again, the authors claim that the overcited findings suggest that motivational signals may affect descending motor pathways to obtain the reward. My feeling is that motivational signals do not affect descending motor pathway "just to obtain reward", but more generally to improve motor performance (e.g. accuracy of the movement, reduction of latencies, increasing of force) modulating, therefore, the motor output.

Response

We thank the Referee for such insightful inputs. We agree your opinion. We modified the text as follow.

Page 6, Line 100-101 (Introduction)

These results suggest that motor output derived from the descending motor pathway may be affected by motivational signals.

3) It is not clear in my opinion the rationale underlying the authors' hypothesis. For instance, in the introduction (line 99) they claim that VM can exert a modulatory impact on cortical activities including the M1. Immediately after they claim they want to investigate direct connections between VM and spinal cord. However, it is not clear for the reader this logical jump. Do they want to investigate direct connections between VM (especially VTA) and the spinal cord because there is functional evidence that suggests that this is the case? Or do they hypothesize the presence of direct connections because the red nucleus which is part of the VM has, in turn, direct connections with the spinal cord? Please, the author should better clarify the rationale of their study.

Response

We do not mention the investigation of the direct connection between the ventral midbrain (VM) and spinal cord in Introduction. We mentioned the investigation of the projections from the VM to the descending motor pathways. To clearly describe our question, we modified the text as follow.

Page 6-7, Line 106-114 (Introduction)

... It is generally accepted that the ventral midbrain (VM), consisting the VTA (termed A10), SNc (A9), and the retrorubral field (RRF; A8), has direct projections to widespread areas of the frontal cortex, including the M1, in primates (Gasper et al., 1992; Williams and Goldman-Rakic, 1998; Zubair et al., 2021) and rodents (Hosp et al., 2011; 2015). Also, the VM can exert a modulatory impact on cortical activities including the M1 (Arsenaults et al., 2014; Kunori et al., 2014; Murriss et al, 2020). However, it is unknown whether the VM can exert modulatory actions on spinal motor outputs via descending motor pathways.

4) With the purpose to demonstrate the anatomical connection between VM and spinal cord, the authors injected into the cervical enlargement respectively BDA and RV tracers. It is clear from Figure 1 that no labeled neurons after BDA injection were found in VTA. However, it is not clear to me, why the authors

excluded direct VM-spinal cord connections after RV injection, at all. We should consider some variables that could affect these results. For example, during RV injections we have a higher number of injected sites (6 tracks for BDA and 13 for RV), a higher injection volume (table 1), and, moreover, different injection sites (C6-T1 for RV injections and C6-C7 for BDA injections). Could these variables affect the different results found after BDA and RV injections? This concern could be justified also from electrophysiological results. Indeed, after M1 inactivation it is still possible to observe some weak muscle responses (Figure 6D, DELTOID) during VM stimulation. Could be this reduced muscle response due to direct connections between VM and spinal cord or between VM and premotor cortices?

Response

In the present experiments, to investigate the direct anatomical connections, we employed a conventional tracer, BDA, which stays in the first-order neurons projecting to the injection site. BDA, even if used with a rather long survival time, is considered to be the most suitable for demonstrating the presence or absence of direct projection.

In contrast, in the case of RV, it is difficult to completely eliminate the possibility of transsynaptic infection even after a short survival time. To precisely determine the appropriate survival time for exclusively detecting the first-order neurons in the current experimental setup (RV injection into the spinal cord), a large number of monkeys are needed. Therefore, we used BDA to confirm whether the VM has direct projections to the spinal cord or not in the current study.

Second, spinal cord injection is a high-risk procedure, and invasion should be kept to a minimum in order to increase the chance of survival. Considering the transport time of the tracer, BDA takes longer time to be transported to the brain from the spinal cord than RV (Survival time was set at 8 weeks for BDA and 84 or 90 hours for RV). Therefore, in order to increase the long-term survival probability of the animals, BDA injection was limited to the C6-C7 spinal segments to minimize the invasion and the injection volume was also reduced.

Third, the C6 and C7 spinal segments contain a large number of motoneurons innervating forelimb muscles. A majority of shoulder, elbow and wrist joint muscles are innervated by the C5-C7 spinal segments (Jenny and Inukai, 1983). As Referee#3 pointed, it is possible that the number of labeled neurons may be affected by the injection volume (3.6 μ l for BDA, 13 or 15.6 μ l for RV) and the injection site (C6-C7 for BDA, C6-T1 for RV). In the present study, the BDA injection was performed into

the C6-C7 spinal segments. However, since there are many forelimb muscles that are innervated by the C6-C7 spinal segments, it is unlikely that differences in the injection volume and site would affect the presence or absence of the label per se.

Lastly, we argue the Referee#3's concern regarding the M1 inactivation experiments. It is difficult to silence all M1 neurons innervating forelimb muscles by the local injection of muscimol (2-mm step). As Referee#3 pointed, if the reduced muscle response [Figure 6D Deltoid (Figure 7D in the revised version)] is due to direct connections between the VM and the spinal cord, it is not reasonable that the BDA results show no labeled neuron at all in the VM. In addition, the example data displayed in Figure 6D (Figure 7D in the revised version) were obtained from Monkey Y. In the case of Monkey Y, we injected muscimol into the wrist and finger areas of the M1 [Figure 6B right (Figure 7B in the revised version)]. Thus, the effect of M1 inactivation on shoulder muscle (Deltoid) might be less compared with that on other muscles. Other possibilities include factors such as the corticospinal tracts arising from the premotor and supplementary motor areas where receive projections from the VM (Gaspar et al., 1992), or VM-brainstem pathways although the existence of projections from the VM to the brainstem-spinal tracts (such as the rubrospinal and reticulospinal pathways) is not clear, or current spreads to the brainstem-spinal tracts. In any case, what the present results show is that the M1 is involved in the generation of muscle activity by VM stimulation.

5) line 486. The authors present data of third-order neurons referring to Purkinje cells. This is an interesting result. However, no mention of its functional role is present in the rest of the manuscript.

Response: As described in Result section (Page 23, Line 493-500), to estimate the order of RV labeling, we investigated the red nucleus–cerebellar circuit by RV labeling of Purkinje cells via the anterior interpositus nucleus (AIP) (Fig. 2A). Because this pathway is already established (Kennedy et al., 1986; Voogd, 2014; Basile et al., 2021). As Referee#2's comments (2), this result is not the main topic of this manuscript. Because this the red nucleus–cerebellar circuit cannot refer to an anatomical linkage between VM and Purkinje cells. We should have embedded Figure 2 into supplemental materials. However, a following sentence is described in information for Authors “The Journal does not usually accept supplemental/supporting material that cannot be included with the body of the manuscript.” Thus, we embed Figure 2 into the main

manuscript.

6) Figure 6C. The authors show the stimulation sites before and during M1 inactivation. Some sites are in VTA, others in SN while others in RRF. I think that could be interesting to show even in a Supplementary figure the muscles' responses relative to each region. Because of our previous knowledge and also the authors' findings, the prediction is that faster and stronger muscle responses during M1 inactivation should be found when the stimulation site is in RRF when compared with VTA stimulations. This would be more informative for the reader and could improve the authors' results.

Response: We thank the Referee#3 for such insightful suggestions. To answer your prediction, we conducted additional analysis comparing regional difference of muscle responses between before and during M1 inactivation (please refer to the following figure).

Figure to Referee#3.

The bar graphs on the left panel show the magnitude of muscle responses before (filled) and during M1 inactivation (transparent) in individual stimulated regions. Regardless of stimulation sites, muscle responses decreased during M1 inactivation (paired t-test). Furthermore, there was no difference in the rate of decrease of muscle responses among stimulation sites (Right panel). This result indicates that M1 contributes to muscle responses induced by the stimulation to all these structures (VTA, SNc, RRF).

Furthermore, we also conducted additional analysis to compare the magnitude of muscle responses among VTA, SNc and RRF obtained from all monkeys used in the present study (Monkey T, D, and Y). Since Monkey Y and Monkey D (left hemisphere stimulation) were used for M1 inactivation study (stimulation parameter: $300 \mu\text{A} \times 3$), data before M1 injection of muscimol or saline are included. In the higher intensity conditions ($500 \mu\text{A} \times 1$ and $500 \mu\text{A} \times 3$), the magnitude of muscle responses induced by the SNc and/or RRF stimulation was larger than that induced by the VTA stimulation (see the following figure). This result indicates that SNc/RRF produce stronger muscle responses compared with the VTA and correspond to the anatomical result showing larger number for RV labeled neurons in SNc and RRF compared with in the VTA. We think this result is in line with Referee#3's expectation. As Referee#3 suggested, this result is informative for the readers. Furthermore, Journal of Physiology does not allow the authors to set the supplementary figures. Therefore, we added this result into the main text as Figure 6A (Figure 6B is from original Figure 5E and modified), and additional description about this result in the Result section (Page 26, Line 547-558). Along with this revise, we change the figure number (Original Figure 6 related to M1 inactivation in Figure 7 in the current version).

Figure 6. Regional differences in muscle responses (A) The effect of current intensity

and number of pulses on responses of multiple muscles induced by the VTA, SNc and RRF stimulations. Data are shown as the mean + SD. For statistical analysis, Kruskal-Wallis tests ($P_{100\mu A \times 1} = 0.145$, $P_{300\mu A \times 1} = 0.180$, $P_{500\mu A \times 1} = 8.54 \times 10^{-4}$, $P_{100\mu A \times 3} = 0.513$, $P_{300\mu A \times 3} = 0.175$, $P_{500\mu A \times 3} = 0.00593$) were performed. Post hoc multiple comparisons with Bonferroni's were performed for $500 \mu A \times 1$ ($P_{VTA \text{ vs } P_{SNc}} = 8.7585 \times 10^{-4}$, $P_{VTA \text{ vs } P_{RRF}} = 1.00$, $P_{SNc \text{ vs } P_{RRF}} = 0.0535$) and $500 \mu A \times 3$ ($P_{VTA \text{ vs } P_{SNc}} = 0.0482$, $P_{VTA \text{ vs } P_{RRF}} = 0.00652$, $P_{SNc \text{ vs } P_{RRF}} = 1$) conditions. *, $P < 0.05$, **, $P < 0.01$, ***, $P < 0.001$. (B) Output effects on proximal and distal muscles. Number of stimulus sites that facilitated only proximal muscles (shoulder, elbow), only distal muscles (wrist, digit, intrinsic hand muscles), or a combination of at least one proximal and at least one distal muscle is divided by the total number of stimulation sites ($100 \mu A \times 1$, $100 \mu A \times 3$ and $300 \mu A \times 1$: 36 sites, $300 \mu A \times 3$: 42 sites, $500 \mu A \times 1$ and $500 \mu A \times 3$: 34 sites). Data were obtained from three monkeys (Monkey T, D and Y). Since Monkey Y and Monkey D (left hemisphere stimulation) were used for M1 inactivation study (stimulation parameter: $300 \mu A \times 3$), data before M1 injection of muscle or saline are included in (A) and (B).

Page 21, Line 439-445 (Methods)

To compare the effect of VM stimulation with different current intensities (100, 300, and $500 \mu A$) and the difference in the magnitude of muscle responses depending on the stimulation sites, we compared MPI among three different stimulus current, number or three regions (VTA, SNc, and RRF) using one-way ANOVA with repeated measures (Kruskal-Wallis test). Post-hoc multiple comparisons were conducted using Bonferroni's correction. Statistical significance was defined as $P < 0.05$.

Page 26, Line 547-564 (Result)

... Comparing the distributions of onset latency ... slow-onset components.

To investigate the difference in the magnitude of muscle responses among the VTA, SNc, and RRF, MPI in each stimulus parameter was compared. In the higher-intensity conditions ($500 \mu A \times 1$ and $500 \mu A \times 3$), the magnitude of muscle responses induced by the SNc and/or RRF stimulation was larger than that induced by the VTA

stimulation (Fig. 6A, Kruskal-Wallis tests: $P_{100\mu\text{A} \times 1} = 0.145$, $P_{300\mu\text{A} \times 1} = 0.180$, $P_{500\mu\text{A} \times 1} = 8.54 \times 10^{-4}$, $P_{100\mu\text{A} \times 3} = 0.513$, $P_{300\mu\text{A} \times 3} = 0.175$, $P_{500\mu\text{A} \times 3} = 0.00593$). Post hoc multiple comparisons with Bonferroni's correction: $P_{\text{VTA vs } P_{\text{SNc}}} = 8.7585 \times 10^{-4}$, $P_{\text{VTA vs } P_{\text{RRF}}} = 1.00$, $P_{\text{SNc vs } P_{\text{RRF}}} = 0.0535$ for $500\mu\text{A} \times 1$ and $P_{\text{VTA vs } P_{\text{SNc}}} = 0.0482$, $P_{\text{VTA vs } P_{\text{RRF}}} = 0.00652$, $P_{\text{SNc vs } P_{\text{RRF}}} = 1$ for $500\mu\text{A} \times 3$). This result indicates that the SNc/RRF produce stronger muscle responses compared with the VTA and corresponds to the anatomical result showing a larger number of RV-labeled neurons in the SNc and RRF compared with in the VTA (Fig. 3). Furthermore, to explore the output preference for the proximal vs. distal muscles in the VTA, SNc, and RRF, we investigated the relationship between the muscle outputs and the stimulus sites. Although there was no clear preference for the proximal (shoulder and elbow) or distal (wrist, digits, and intrinsic hand) muscles in the stimulus effects of VTA, SNc, and RRF, the RRF appeared to have divergent outputs to both the proximal and the distal muscles, as compared to the VTA and SNc (Fig. 6B).

Page 34, Line 740-743 (Discussion)

... Further studies are needed to elucidate the functional roles of the VM–spinal pathway during motor performance, motor learning, and rehabilitation from motor impairments and, also, the functional differences among the VTA, SNc, and RRF in motivated behavior and motor control.

END OF COMMENTS

Dear Dr Nishimura,

Re: JP-RP-2021-282429R1 "A multisynaptic pathway from the ventral midbrain toward spinal motoneurons in monkeys" by Michiaki Suzuki, Ken-ichi Inoue, Hiroshi Nakagawa, Hiroaki Ishida, Kenta Kobayashi, Tadashi Isa, Masahiko Takada, and Yukio Nishimura

Thank you for submitting your revised Research Article to The Journal of Physiology. It has been assessed by the original Reviewing Editor and Referees and has been well received. Some final revisions have been requested.

The reports are copied at the end of this email. Please address all of the points and incorporate all requested revisions, or explain in your Response to Referees why a change has not been made.

NEW POLICY: In order to improve the transparency of its peer review process The Journal of Physiology publishes online as supporting information the peer review history of all articles accepted for publication. Readers will have access to decision letters, including all Editors' comments and referee reports, for each version of the manuscript and any author responses to peer review comments. Referees can decide whether or not they wish to be named on the peer review history document.

I hope you will find the comments helpful and have no difficulty returning revisions within 2 weeks.

If you need to check to make sure that your Methods section conforms to the principles of UK regulations, you may wish to refer to Grundy (2015):

Grundy (2015) J. Physiol. 2015 Jun 15;593(12):2547-9 <https://doi.org/10.1113/JP270818>

Your revised manuscript should be submitted online using the links in Author Tasks Link Not Available. This link is to the Corresponding Author's own account, if this will cause any problems when submitting the revised version please contact us.

The image files from the previous version are retained on the system. Please ensure you replace or remove any files that have been revised.

REVISION CHECKLIST:

- Summary data must be reported as mean {plus minus} SD or 95% confidence interval
- All table and figure legends with summary data must include the statistical test used in the table/figure and sample size
- Figures with summary data bars must include individual data points, or box whisker plots when $n > 30$.
- Article file, including any tables and figure legends, must be in an editable format (eg Word)
- Upload each figure as a separate high quality file
- Upload a full Response to Referees, including a response to any Senior and Reviewing Editor Comments;
- Upload a copy of the manuscript with the changes highlighted.

- A potential 'Cover Art' file for consideration as the Issue's cover image;
- Appropriate Supporting Information (Video, audio or data set https://jp.msubmit.net/cgi-bin/main.plex?form_type=display_requirements#supp).

To create your 'Response to Referees' copy all the reports, including any comments from the Senior and Reviewing Editors, into a Word, or similar, file and respond to each point in colour or CAPITALS and upload this when you submit your revision.

I look forward to receiving your revised submission.

If you have any queries please reply to this email and the Peer Review Coordinator will be pleased to advise.

If revision is not possible, or if you cannot respond to the requests for change, contact us by return email as soon as

possible, giving reasons for the difficulties. Withdrawal of the manuscript may be necessary in these circumstances, and instruction will be given on how to proceed. Please note that a paper must be withdrawn before it can be submitted to another journal. If any issues remain unresolved please contact the Publications Office at jphysiol@physoc.org

If you would like help with English language editing, or other article preparation support, Wiley Editing Services offers expert help with English Language Editing, as well as translation, manuscript formatting, and figure formatting at www.wileyauthors.com/eeo/preparation. You can also check out our resources for Preparing Your Article for general guidance about writing and preparing your manuscript at www.wileyauthors.com/eeo/prepresources.

Yours sincerely,

Richard Carson
Senior Editor
The Journal of Physiology

REQUIRED ITEMS:

-You must start the Methods section with a paragraph headed Ethical Approval. A detailed explanation of journal policy and regulations on animal experimentation is given in Principles and standards for reporting animal experiments in The Journal of Physiology and Experimental Physiology by David Grundy J Physiol, 593: 2547-2549. doi:10.1113/JP270818.). A checklist outlining these requirements and detailing the information that must be provided in the paper can be found at: <https://physoc.onlinelibrary.wiley.com/hub/animal-experiments>. Authors should confirm in their Methods section that their experiments were carried out according to the guidelines laid down by their institution's animal welfare committee, and conform to the principles and regulations as described in the Editorial by Grundy (2015). The Methods section must contain details of the anaesthetic regime: anaesthetic used, dose and route of administration and method of killing the experimental animals.

-The Journal of Physiology funds authors of provisionally accepted papers to use the premium BioRender site to create high resolution schematic figures. Follow this link and enter your details and the manuscript number to create and download figures. Upload these as the figure files for your revised submission. If you choose not to take up this offer we require figures to be of similar quality and resolution. If you are opting out of this service to authors, state this in the Comments section on the Detailed Information page of the submission form.

-Papers must comply with the Statistics Policy https://jp.msubmit.net/cgi-bin/main.plex?form_type=display_requirements#statistics

In summary:

-If n {less than or equal to} 30, all data points must be plotted in the figure in a way that reveals their range and distribution. A bar graph with data points overlaid, a box and whisker plot or a violin plot (preferably with data points included) are acceptable formats.

-If $n > 30$, then the entire raw dataset must be made available either as supporting information, or hosted on a not-for-profit repository e.g. FigShare, with access details provided in the manuscript.

-' n ' clearly defined (e.g. x cells from y slices in z animals) in the Methods. Authors should be mindful of pseudoreplication.

-All relevant ' n ' values must be clearly stated in the main text, figures and tables, and the Statistical Summary Document (required upon revision)

-The most appropriate summary statistic (e.g. mean or median and standard deviation) must be used. Standard Error of the Mean (SEM) alone is not permitted.

-Exact p values must be stated. Authors must not use 'greater than' or 'less than'. Exact p values must be stated to three significant figures even when 'no statistical significance' is claimed.

-Statistics Summary Document completed appropriately upon revision

EDITOR COMMENTS

Reviewing Editor:

Please revised surgical procedures as the Animal Ethics Editor suggested.

REFeree COMMENTS

Referee #1:

Thank you for addressing the points raised previously. Some additional important information is required.

You have stated:

Page 9, Line 165-188

"Surgery for viral injections into the spinal cord. The surgeries described below were performed under general anesthesia initiated by ketamine (10 mg/kg, i.m.) plus xylazine

(1 mg/kg, i.m.) and maintained with sodium pentobarbital (20 mg/kg, i.v.) or 1-1.5% isoflurane. The depth of anesthesia was confirmed by pain response. During anesthesia, the animal's vital signs, such as respiratory/circulatory parameters and body temperature, were carefully monitored".

This should be written to confirm the absence of a pain response under general anaesthesia. The authors need to be explicit in stating what was assessed. Respiratory and cardiovascular parameters were measured. What parameters were assessed to determine an adequate depth of anaesthesia? For example, where appropriate, the authors could comment on the absence of the pedal withdrawal reflex and/or absence of pressor response or tachypnoea to noxious stimulus etc. I appreciate that this issue was carefully addressed by the authors in the study. It is important that these steps are expressed in the published article to give readers an assurance of the careful approach taken to ensure animal welfare during surgical procedures.

Referee #2:

The revised manuscript has satisfied my comments. There is no further feedback from my side.

Referee #3:

The authors have responded to all concerns and doubts of the reviewer.

END OF COMMENTS

1st Confidential Review

24-Nov-2021

We greatly appreciate the editors and the three referees for their careful reading of our manuscript and helpful comments on our manuscript. We addressed and incorporated all the comments in the revised manuscript. We have made two changes that are marked in blue. Below we list our responses to each of the editors' and referees' comments.

EDITOR COMMENTS

Reviewing Editor:

Please revised surgical procedures as the Animal Ethics Editor suggested.

Response

We added detail descriptions about surgical procedures suggested by the Animal Ethics Editor (Referee#1).

REFEREE COMMENTS

Referee #1:

Thank you for addressing the points raised previously. Some additional important information is required.

You have stated:

Page 9, Line 165-188

"Surgery for viral injections into the spinal cord. The surgeries described below were performed under general anesthesia initiated by ketamine (10 mg/kg, i.m.) plus xylazine (1 mg/kg, i.m.) and maintained with sodium pentobarbital (20 mg/kg, i.v.) or 1-1.5% isoflurane. The depth of anesthesia was confirmed by pain response. During anesthesia, the animal's vital signs, such as respiratory/circulatory parameters and body temperature, were carefully monitored".

This should be written to confirm the absence of a pain response under general

anaesthesia. The authors need to be explicit in stating what was assessed. Respiratory and cardiovascular parameters were measured. What parameters were assessed to determine an adequate depth of anaesthesia? For example, where appropriate, the authors could comment on the absence of the pedal withdrawal reflex and/or absence of pressor response or tachypnoea to noxious stimulus etc. I appreciate that this issue was carefully addressed by the authors in the study. It is important that these steps are expressed in the published article to give readers an assurance of the careful approach taken to ensure animal welfare during surgical procedures.

Response

We added additional descriptions about each surgery.

Page 9, Line 165-173

Surgery for viral injections into the spinal cord. The surgeries described below were performed under general anesthesia initiated by ketamine (10 mg/kg, i.m.) plus xylazine (1 mg/kg, i.m.) and maintained with sodium pentobarbital (20 mg/kg, i.v.) or 1-1.5% isoflurane. The depth of anesthesia was confirmed by pain response. During **surgery**, the animal's vital signs, such as respiratory/circulatory parameters (**respiratory rate, inspiratory carbon dioxide concentration, saturation of percutaneous oxygen and heart rate**) and body temperature, were carefully monitored. **The absence of tachypnea and tachycardia to noxious stimulus was also used to verify the level of anesthesia.** All surgical procedures were performed in sterile conditions....

Page 13, Line 272-279

Surgery for chronic implants. The surgeries described below were performed under general anesthesia initiated by ketamine (10 mg/kg, i.m.) plus xylazine (1 mg/kg, i.m.) and maintained with 1-1.5% isoflurane. The depth of anesthesia was confirmed by pain response. During **surgery**, the animal's vital signs, such as respiratory/circulatory parameters (**respiratory rate, inspiratory carbon dioxide concentration, saturation of percutaneous oxygen and heart rate**) and body temperature, were carefully monitored. **The absence of tachypnea and tachycardia to noxious stimulus was also used to verify the level of anesthesia.** All surgical procedures were performed in sterile conditions. ...

Referee #2:

The revised manuscript has satisfied my comments. There is no further feedback from my side.

Referee #3:

The authors have responded to all concerns and doubts of the reviewer.

END OF COMMENTS

Dear Dr Nishimura,

Re: JP-RP-2021-282429R2 "A multisynaptic pathway from the ventral midbrain toward spinal motoneurons in monkeys" by Michiaki Suzuki, Ken-ichi Inoue, Hiroshi Nakagawa, Hiroaki Ishida, Kenta Kobayashi, Tadashi Isa, Masahiko Takada, and Yukio Nishimura

Thank you for submitting your revised Research Article to The Journal of Physiology. It has been assessed by the original Reviewing Editor and Referees and has been well received. Some final revisions have been requested.

The reports are copied at the end of this email. Please address all of the points and incorporate all requested revisions, or explain in your Response to Referees why a change has not been made.

NEW POLICY: In order to improve the transparency of its peer review process The Journal of Physiology publishes online as supporting information the peer review history of all articles accepted for publication. Readers will have access to decision letters, including all Editors' comments and referee reports, for each version of the manuscript and any author responses to peer review comments. Referees can decide whether or not they wish to be named on the peer review history document.

I hope you will find the comments helpful and have no difficulty returning revisions within one week.

If you need to check to make sure that your Methods section conforms to the principles of UK regulations, you may wish to refer to Grundy (2015):

Grundy (2015) J. Physiol. 2015 Jun 15;593(12):2547-9 <https://doi.org/10.1113/JP270818>

Your revised manuscript should be submitted online using the links in Author Tasks Link Not Available. This link is to the Corresponding Author's own account, if this will cause any problems when submitting the revised version please contact us.

The image files from the previous version are retained on the system. Please ensure you replace or remove any files that have been revised.

REVISION CHECKLIST:

- Summary data must be reported as mean {plus minus} SD or 95% confidence interval
- All table and figure legends with summary data must include the statistical test used in the table/figure and sample size
- Figures with summary data bars must include individual data points, or box whisker plots when $n > 30$.
- Article file, including any tables and figure legends, must be in an editable format (eg Word)
- Upload each figure as a separate high quality file
- Upload a full Response to Referees, including a response to any Senior and Reviewing Editor Comments;
- Upload a copy of the manuscript with the changes highlighted.

- A potential 'Cover Art' file for consideration as the Issue's cover image;
- Appropriate Supporting Information (Video, audio or data set https://jp.msubmit.net/cgi-bin/main.plex?form_type=display_requirements#supp).

To create your 'Response to Referees' copy all the reports, including any comments from the Senior and Reviewing Editors, into a Word, or similar, file and respond to each point in colour or CAPITALS and upload this when you submit your revision.

I look forward to receiving your revised submission.

If you have any queries please reply to this email and the Peer Review Coordinator will be pleased to advise.

If revision is not possible, or if you cannot respond to the requests for change, contact us by return email as soon as

possible, giving reasons for the difficulties. Withdrawal of the manuscript may be necessary in these circumstances, and instruction will be given on how to proceed. Please note that a paper must be withdrawn before it can be submitted to another journal. If any issues remain unresolved please contact the Publications Office at jphysiol@physoc.org

If you would like help with English language editing, or other article preparation support, Wiley Editing Services offers expert help with English Language Editing, as well as translation, manuscript formatting, and figure formatting at www.wileyauthors.com/eeo/preparation. You can also check out our resources for Preparing Your Article for general guidance about writing and preparing your manuscript at www.wileyauthors.com/eeo/prepresources.

Yours sincerely,

Richard Carson
Senior Editor
The Journal of Physiology

REQUIRED ITEMS:

-You must start the Methods section with a paragraph headed Ethical Approval. A detailed explanation of journal policy and regulations on animal experimentation is given in Principles and standards for reporting animal experiments in The Journal of Physiology and Experimental Physiology by David Grundy J Physiol, 593: 2547-2549. doi:10.1113/JP270818.). A checklist outlining these requirements and detailing the information that must be provided in the paper can be found at: <https://physoc.onlinelibrary.wiley.com/hub/animal-experiments>. Authors should confirm in their Methods section that their experiments were carried out according to the guidelines laid down by their institution's animal welfare committee, and conform to the principles and regulations as described in the Editorial by Grundy (2015). The Methods section must contain details of the anaesthetic regime: anaesthetic used, dose and route of administration and method of killing the experimental animals.

-The Journal of Physiology funds authors of provisionally accepted papers to use the premium BioRender site to create high resolution schematic figures. Follow this link and enter your details and the manuscript number to create and download figures. Upload these as the figure files for your revised submission. If you choose not to take up this offer we require figures to be of similar quality and resolution. If you are opting out of this service to authors, state this in the Comments section on the Detailed Information page of the submission form.

EDITOR COMMENTS

Reviewing Editor:

The authors have addressed all the issues raised by the reviewers except for one remaining point below.

REFEREE COMMENTS

Referee #1:

Thank you for the revisions made. There is one remaining point. Assessment of heart rate during noxious stimulation is complicated by the potential for bradycardia arising from the baroreflex, that is, a reflex slowing of the heart caused by a rise in blood pressure. Therefore an absence of tachycardia is not in-and-of itself fully descriptive. Where appropriate, the authors should state that there was no evidence of tachycardia during surgical procedures and no major deviation in heart rate during noxious stimulation. The comment on tachypnoea can remain.

Cardiorespiratory assessments are very useful during surgical and experimental procedures, but brainstem-dependent reflexes can remain intact during anaesthesia. The absence of gross movement (pedal reflex) and other assessments (eg corneal reflex) are commonly observed and reported. Where appropriate, I encourage the authors to include such a statement.

END OF COMMENTS

We greatly appreciate the editor and the referee for their careful reading of our manuscript and helpful comments on our manuscript. We addressed and incorporated all the comments in the revised manuscript. We have made two changes that are marked in blue. Below we list our responses to each of the editor's and referee's comments.

EDITOR COMMENTS

Reviewing Editor:

The authors have addressed all the issues raised by the reviewers except for one remaining point below.

Response

We added detail descriptions about surgical procedures suggested by the Referee#1.

REFEREE COMMENTS

Referee #1:

Thank you for the revisions made. There is one remaining point. Assessment of heart rate during noxious stimulation is complicated by the potential for bradycardia arising from the baroreflex, that is, a reflex slowing of the heart caused by a rise in blood pressure. Therefore an absence of tachycardia is not in-and-of itself fully descriptive. Where appropriate, the authors should state that there was no evidence of tachycardia during surgical procedures and no major deviation in heart rate during noxious stimulation. The comment on tachypnoea can remain.

Cardiorespiratory assessments are very useful during surgical and experimental procedures, but brainstem-dependent reflexes can remain intact during anaesthesia. The absence of gross movement (pedal reflex) and other assessments (eg corneal reflex) are commonly observed and reported. Where appropriate, I encourage the authors to include such a statement.

Response

We added additional descriptions about each surgery as follows.

Page 8, Line165-174

Surgery for viral injections into the spinal cord. The surgeries described below were performed under general anesthesia initiated by ketamine (10 mg/kg, i.m.) plus xylazine (1 mg/kg, i.m.) and maintained with sodium pentobarbital (20 mg/kg, i.v.) or 1-1.5% isoflurane. The depth of anesthesia was confirmed by pain response. During surgery, the animal's vital signs, such as respiratory/circulatory parameters (respiratory rate, inspiratory carbon dioxide concentration, saturation of percutaneous oxygen and heart rate) and body temperature, were carefully monitored. There was no evidence for tachycardia or tachypnea during surgical procedures, nor major deviation in the heart or respiratory rate in response to noxious stimuli. The absence of reflexive movements to noxious stimuli and corneal reflex was also used to verify the level of anesthesia. ...

Page 13, Line 274-283

Surgery for chronic implants. The surgeries described below were performed under general anesthesia initiated by ketamine (10 mg/kg, i.m.) plus xylazine (1 mg/kg, i.m.) and maintained with 1-1.5% isoflurane. The depth of anesthesia was confirmed by pain response. During surgery, the animal's vital signs, such as respiratory/circulatory parameters (respiratory rate, inspiratory carbon dioxide concentration, saturation of percutaneous oxygen and heart rate) and body temperature, were carefully monitored. There was no evidence for tachycardia or tachypnea during surgical procedures, nor major deviation in the heart or respiratory rate in response to noxious stimuli. The absence of reflexive movements to noxious stimuli and corneal reflex was also used to verify the level of anesthesia....

END OF COMMENTS

Dear Dr Nishimura,

Re: JP-RP-2022-282429R3 "A multisynaptic pathway from the ventral midbrain toward spinal motoneurons in monkeys" by Michiaki Suzuki, Ken-ichi Inoue, Hiroshi Nakagawa, Hiroaki Ishida, Kenta Kobayashi, Tadashi Isa, Masahiko Takada, and Yukio Nishimura

I am pleased to tell you that your paper has been accepted for publication in The Journal of Physiology, subject to any modifications to the text and/or satisfactory clarification of the Methods section that may be required by the Journal Office to conform to House rules.

NEW POLICY: In order to improve the transparency of its peer review process The Journal of Physiology publishes online as supporting information the peer review history of all articles accepted for publication. Readers will have access to decision letters, including all Editors' comments and referee reports, for each version of the manuscript and any author responses to peer review comments. Referees can decide whether or not they wish to be named on the peer review history document.

The last Word version of the paper submitted will be used by the Production Editors to prepare your proof. When this is ready you will receive an email containing a link to Wiley's Online Proofing System. The proof should be checked and corrected as quickly as possible.

Authors should note that it is too late at this point to offer corrections prior to proofing. Major corrections at proof stage, such as changes to figures, will be referred to the Reviewing Editor for approval before they can be incorporated. Only minor changes, such as to style and consistency, should be made a proof stage. Changes that need to be made after proof stage will usually require a formal correction notice.

All queries at proof stage should be sent to TJP@wiley.com

The accepted version of the manuscript will be published online, prior to copy editing, in the Accepted Articles section.

Are you on Twitter? Once your paper is online, why not share your achievement with your followers. Please tag The Journal (@jphysiol) in any tweets and we will share your accepted paper with our 22,000+ followers!

Yours sincerely,

Richard Carson
Senior Editor
The Journal of Physiology

P.S. - You can help your research get the attention it deserves! Check out Wiley's free Promotion Guide for best-practice recommendations for promoting your work at www.wileyauthors.com/eeo/guide. And learn more about Wiley Editing Services which offers professional video, design, and writing services to create shareable video abstracts, infographics, conference posters, lay summaries, and research news stories for your research at www.wileyauthors.com/eeo/promotion.

* IMPORTANT NOTICE ABOUT OPEN ACCESS *

Information about Open Access policies can be found here <https://physoc.onlinelibrary.wiley.com/hub/access-policies>

To assist authors whose funding agencies mandate public access to published research findings sooner than 12 months after publication The Journal of Physiology allows authors to pay an open access (OA) fee to have their papers made freely available immediately on publication.

You will receive an email from Wiley with details on how to register or log-in to Wiley Authors Services where you will be able to place an OnlineOpen order.

You can check if your funder or institution has a Wiley Open Access Account here <https://authorservices.wiley.com/author-resources/Journal-Authors/licensing-and-open-access/open-access/author-compliance-tool.html>

Your article will be made Open Access upon publication, or as soon as payment is received.

If you wish to put your paper on an OA website such as PMC or UKPMC or your institutional repository within 12 months of publication you must pay the open access fee, which covers the cost of publication.

OnlineOpen articles are deposited in PubMed Central (PMC) and PMC mirror sites. Authors of OnlineOpen articles are permitted to post the final, published PDF of their article on a website, institutional repository, or other free public server, immediately on publication.

Note to NIH-funded authors: The Journal of Physiology is published on PMC 12 months after publication, NIH-funded authors DO NOT NEED to pay to publish and DO NOT NEED to post their accepted papers on PMC.

EDITOR COMMENTS

Reviewing Editor:

Thank you for revising the whole manuscript.

REFeree COMMENTS

Referee #1:

Thank you for the revision. There are no further issues.

3rd Confidential Review

07-Jan-2022